# Identification of osteogenic progenitor cell-targeted peptides that augment bone formation

Min Jiang[1,2], Ruiwu Liu[3], Lixian Liu[1], Alexander Kot [1], Xueping Liu[1], Wenwu Xiao[3], Junjing Jia[1], Yuanpei Li [3], Kit S. Lam[3] & Wei Yao [1✉]

Activation and migration of endogenous mesenchymal stromal cells (MSCs) are critical for bone regeneration. Here, we report a combinational peptide screening strategy for rapid discovery of ligands that not only bind strongly to osteogenic progenitor cells (OPCs) but also stimulate osteogenic cell Akt signaling in those OPCs. Two lead compounds are discovered, YLL3 and YLL8, both of which increase osteoprogenitor osteogenic differentiation *in vitro*. When given to normal or osteopenic mice, the compounds increase mineral apposition rate, bone formation, bone mass, and bone strength, as well as expedite fracture repair through stimulated endogenous osteogenesis. When covalently conjugated to alendronate, YLLs acquire an additional function resulting in a "tri-functional" compound that: (i) binds to OPCs, (ii) targets bone, and (iii) induces "pro-survival" signal. These bone-targeted, osteogenic peptides are well suited for current tissue-specific therapeutic paradigms to augment the endogenous osteogenic cells for bone regeneration and the treatment of bone loss.

---

[1] Center for Musculoskeletal Health, Department of Internal Medicine, The University of California at Davis Medical Center, Sacramento, CA 95817, USA. [2] Shanghai Key Laboratory for Prevention and Treatment of Bone and Joint Diseases, Shanghai Institute of Traumatology and Orthopaedics, Ruijin Hospital, Shanghai Jiao Tong University School of Medicine, 197 Ruijin 2nd Road, 200025 Shanghai, China. [3] Department of Biochemistry and Molecular Medicine, The University of California at Davis Medical Center, Sacramento, CA 95817, USA. ✉email: yao@ucdavis.edu

Osteoporosis is a disease of increased bone fragility that initially results from estrogen deficiency and aging. It is a significant public health problem with nearly 50% of all Caucasian women and 25% of Caucasian men at risk for an osteoporotic fracture in their lifetime (Publication from National Osteoporosis Foundation). With the aging segment of our population growing rapidly worldwide, osteoporosis has become a significant health concern. More than 2 million osteoporotic fractures occur annually in the US, with 27% vertebral and 14% hip fractures[1]. Aging is associated with a reduction in bone marrow skeletal osteoprogenitor cells and the proper micro-environment to support these osteoprogenitors to differentiate into osteoblasts to form bone[2,3]. Currently, most treatments for osteoporosis reduce bone loss by decreasing osteoclastic bone resorption, which prevents further breakdown of bone. However, anti-resorptive medications do not restore the lost bone structure. Therapeutic modalities that target bone formation by increasing the number of or the activity of osteoblasts may be a more attractive approach to enhance bone formation and promote bone regeneration. Current medications that enhance bone formation include hPTH (1–34) (Teriparatide), PTHrp (Abaloparatide), and Evenity (Romosozumab)[4–7], all of which are FDA approved for the treatment of osteoporosis[8,9]. Teriparatide and Abaloparatide treatment are limited to 2 years with a boxed warning of risk for osteosarcomas. Neither Teriparatide nor Abaloparatide are recommended for patients at a higher risk for osteosarcoma, who have a history of irradiation, or who have primary or secondary hyperparathyroidism. These anabolic agents can reduce the risk of vertebral fractures more than 50%, but they remain less effective at reducing hip fractures[6,10]. Therefore, there is still a need to search for bone-specific osteogenic anabolic drugs for the treatment of osteoporosis.

The activation and migration of endogenous mesenchymal stromal cells (MSCs) are critical for bone regeneration. MSCs are progenitor cells that are capable of differentiating into osteoblasts and chondrocytes[11]. Lam and colleagues[12,13] previously reported the discovery of LLP2A, a high-affinity peptidomimetic ligand against activated α4β1 integrin of lymphoid cancer, later it was found that this compound bound strongly to MSCs. Conjugation of LLP2A to alendronate, a bisphosphonate, resulted in (i) efficient immobilizing of LLP2A to the bone, (ii) mobilization of circulating MSCs to the bone, and (iii) promotion of bone growth in osteopenic animals[14–17]. In this study, we aim to develop a strategy to identify ligands that could mobilize not only osteogenic progenitor cells (OPCs) to the bone but also induce intracellular pro-survival signaling, i.e., activation of Akt signaling of the OPCs upon cell binding. To achieve this, we use osterix+ OPC cells as a living probe to screen a focused one-bead one-compound (OBOC) combinatorial library and use in-cell immunohistochemistry to stain bead-bound cells for Akt phosphorylation. Through a two-step screening process, we identify two leading osteogenic peptides, YLL3 and YLL8, that can increase in vitro osteogenic differentiation of the OPCs derived from both human and murine MSCs. Treatment with these compounds increases mineral apposition rate, bone formation, bone mass, and bone strength in normal and osteopenic mice, as well as expedites fracture repair through enhanced endogenous osteogenesis. Furthermore, two doses of YLL8-Alendronate (YLL8-Aln) conjugate increase trabecular bone mass and strength to similar levels as daily injections of the YLL8 peptide or PTH for 28 days. These bone cell-targeted, osteogenic peptides are well suited for current tissue-specific therapeutic paradigms to augment the endogenous osteogenic cells for bone regeneration/ bone growth.

## Results

**OBOC library screens for OPC-targeted anabolic peptides**. The two known binding peptide motifs for α4β1 integrin are "LDV"[18] and "QIDS"[19]. MSCs and OPCs are known to express α4β1 integrin[16,17,20,21]. We, therefore, synthesized a focused OBOC combinatorial peptidomimetic library biased towards the "LDV" and "QIDS" motifs and N-terminally capped with 4-[(N′-2-methyl phenyl)ureido]phenylacetic acid, a known pharmacophore, for high-affinity ligands against α4β1 integrin[22] (Supplementary Fig. 1). The OBOC library had four diversities with the permutation of 185,760, and 43, 30, 8, and 18 amino acids at position $X_1$, $X_2$, $X_3$, and $X_4$, respectively. The library was then screened with OPCs for cell binding and Akt cell signaling. To prepare OPCs, bone marrow stromal cells (BMSCs) were obtained from humans and mice, maintained in mesenchymal maintenance medium for 2 weeks (~3 passages), and then switched to osteogenic medium for 3 days to stimulate the MSCs towards osteogenic differentiation to the OPCs[16,17,23,24]. After incubating OPCs with the focused OBOC library for 1 h, the bead-bound cells were then briefly fixed with formaldehyde, permeated and stained for phosphorylated Akt, a signaling transduction pathway that promotes cell survival and growth in response to extracellular signals[25]. Beads coated with green fluorescent cells indicated that the ligand displayed on the bead surface-bound OPCs and activated p-Akt upon interaction with cell surface receptor, in this case, α4β1 integrin (Fig. 1a, stars). The positive beads (Fig. 1a, stars) were manually picked under a dissecting microscope and subjected to microsequencing with automated Edman degradation[26]. A total of 22 positive beads were identified and micro-sequenced. These 22 ligands were resynthesized on the surface of the beads but with a nitrotyrosine (fluorescence quencher) displayed at the interior of the beads, such that autofluorescence from the beads could be quenched to facilitate the next stage of testing. Small samples from each of these 22 positive beads were incubated with OPCs obtained from osterix-mCherry reporter mice. In these mice, Osterix is expressed by osteochondral progenitors and is accepted as an early marker for osteoblast maturation[27,28]. The positive beads from the second round of screening were confirmed by their binding affinity toward osterix+OPCs (Fig. 1b). Also, p-Akt activation in these cells upon binding to beads displaying the peptides YLL3 or YLL8 was confirmed (Fig. 1c). We noted that the cells that did not bind to the beads were Akt negative (Fig. 1c). Of the 22 ligands, we selected two peptides, YLL3 and YLL8 (Fig. 1d and Supplementary Figs. 1, 2a; Tables S1-4), as lead compounds for further characterization and development, based on their high-binding affinity to osterix cells, activation of p-Akt, as well as their osteogenic effects in vitro. Importantly, beads displaying YLL3 and YLL8 also showed a low affinity to lymphocytes (Supplementary Fig. 2b), making them more tissue-specific for in vivo applications.

**OPC-targeting peptides are osteogenic in vitro**. We first evaluated the effects of YLL3 and YLL8 on osteogenesis in vitro by measuring osteogenic differentiation (alkaline phosphatase activity; ALP) at day 7 or 10 and osteoblast maturation (alizarin red staining; AR) at day 21 in OPCs derived from humans or mice. In human-derived OPCs, we found that compared to beads displaying a scrambled peptide (Con), beads displaying YLL3 or YLL8 peptide increased ALP levels at day 7 and AR levels at day 21 (Fig. 2a). The beads displaying YLL3 or YLL8 peptides had a high affinity towards human OPCs (Fig. 2b) and cells surrounding those beads were highly positive for ALP at day 10 (Fig. 2c, black arrows). Increased ALP levels, a marker for

**Fig. 1 Schematic for the discovery of osteogenic-specific peptides. a** Bone marrow stromal cells (BMSCs) were obtained from mice and maintained in mesenchymal maintenance medium for 2 weeks (~3 passages), switched to osteogenic medium for 3 to 5 days, then incubated with a focused integrin library for 1 h. These bead-bound osteoprogenitor cells (OPCs) were then fixed and stained for phosphorylated Akt (p-Akt). Positive beads were identified as displaying both cell binding to the beads and positive p-Akt expression in the cells bound to the beads. **b** Positive beads from the first screen were identified, sequenced, and resynthesized. The beads were incubated with osterix+ red cells to identify beads with a high affinity towards osterix+ cells semi-quantitatively. **c** Positive beads from the second screen were resynthesized and re-incubated with OPCs to confirm p-Akt activation. P-Akt was stained green. Some cells on beads were positive for p-Akt. Cells not attached to beads were p-Akt negative. **d** From the described screening process, YLL3 and YLL8 were selected for decoding and resynthesized for further characterization. All the experiments were repeated at least three times. Scale bar = 100 µm.

osteogenic differentiation, were confirmed by ALP staining at day 10 (Fig. 2e). Similarly, in mouse-derived OPCs, beads displaying YLL3 or YLL8 peptide increased ALP levels compared to beads displaying a scrambled peptide (Control; Fig. 2e) and induced higher ALP levels (Fig. 2f) and mineralization nodule formation (Fig. 2g). The cells surrounded the YLL3 and YLL8-displaying beads and deposited mineral (Fig. 2g). These results were confirmed by directly adding the YLL3 or YLL8 peptides to the osteoblast differentiation culture (Fig. 2h). Moreover, the cell viability remained stable when the human MSCs were cultured with up to $2 \times 10^{-6}$ M YLL3 or YLL8 for 3 days (Supplementary Fig. 3a, b).

Since our primary functional screening mechanism was focused on the potential activation of the Akt pathway, we next confirmed the activation of the Akt pathway by incubating the peptides ($6 \times 10^{-8}$ M) with mouse-derived OPCs in osteogenic medium for 3 days. Human PTH (1–34) ($6 \times 10^{-8}$ M) was used as a positive control (Fig. 3a–b). As it was shown in Fig. 3a, compared to the control and YLL3, YLL8 had a very similar profile to that of PTH in activating members in the Akt signaling pathway (Fig. 3a, rectangle boxes), including phosphorylation of Akt, Phosphoinositide-dependent kinase 1 (PDK1), extracellular signaling-regulated kinase (ERK1/2), p53, BCL2-Associated Agonist of Cell Death (BAD), and RIK1 (Fig. 3c). YLL3 activated p53 and Akt (Fig. 3c). We also repeated similar studies using human-derived OPCs incubated with $10^{-9}$ M to $10^{-6}$ M YLL3 or YLL8 on p-Akt activation and semi-quantitated by western blots (Fig. 3d, e). These results confirmed that these OPC-targeting peptides increased osteoblast differentiation and maturation through promoting signaling pathway ways for cell growth or pro-surviving mechanism through activation of p-Akt.

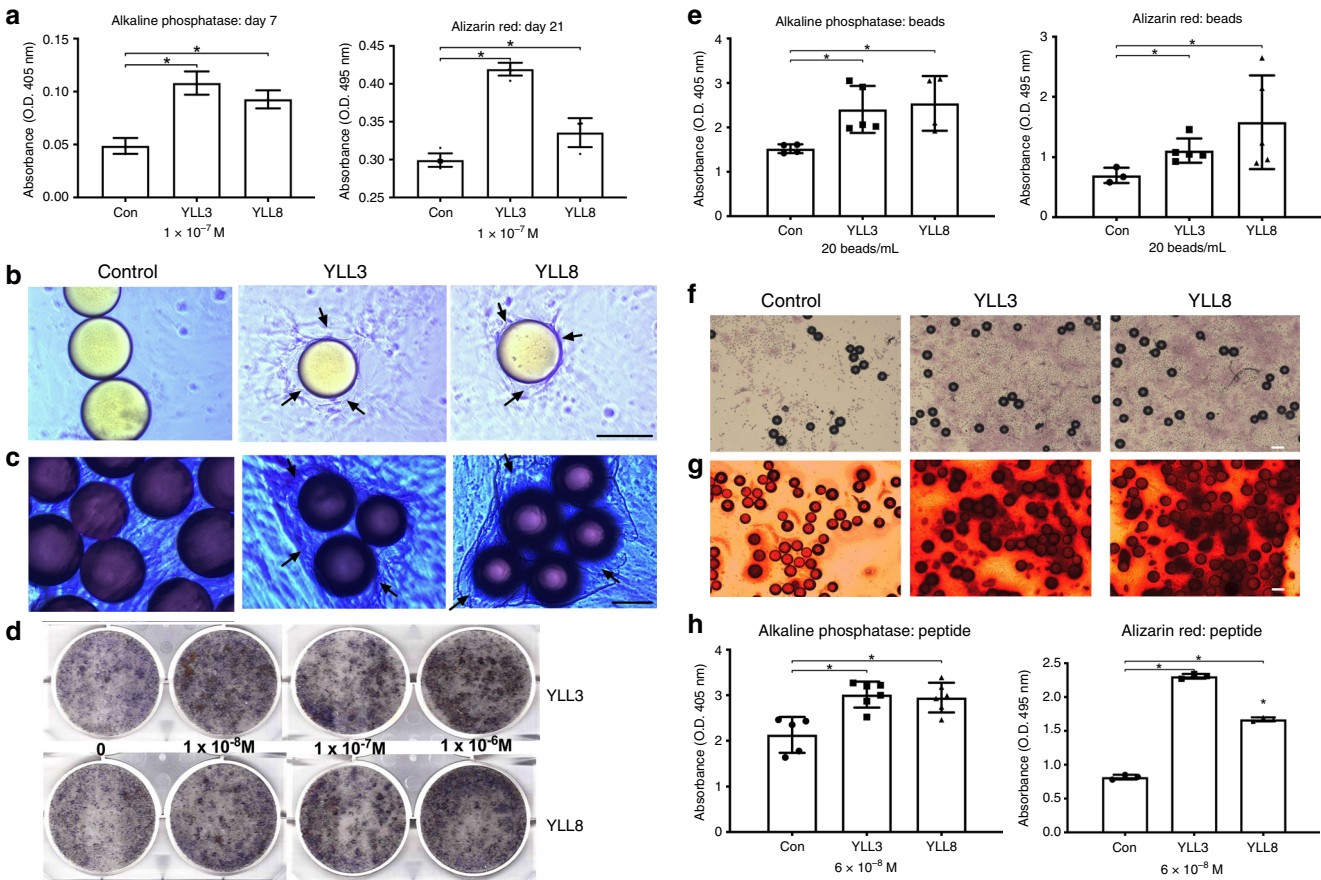

**Fig. 2 Osteogenic peptides YLL3 and YLL8-binding affinities towards osteoprogenitor cells and in vitro osteogenic differentiation property.** BMSCs obtained from mouse or human were plated on collagen-I coated wells and incubated with autofluorescence-quenched beads displaying YLL3 or YLL8 in osteogenic media. **a** Quantitative measurement of alkaline phosphatase (ALP) at day 7 and alizarin red (AR) levels at day 21 in human-derived cell cultures treated with YLL3 or YLL8. Columns represented individual values in each group and the error bars represented standard deviations. $N = 3$ biologically independent samples/group. *$p < 0.05$ between the treated groups versus control (Con) using one-way ANOVA followed by Dunnett's multiple comparison post hoc test. **b** Representative images of binding affinities with human-derived MSCs after co-cultured with beads displaying YLL3 or YLL8 for 12 h. Scale bar = 100 µm. **c** Representative images of ALP staining in human-derived MSCs cultured with beads displaying YLL3 or YLL8 for 14 days. Scale bar = 100 µm. **d** Human-derived MSCs were cultured with YLL3, and YLL8 peptides at indicated concentration in osteogenic media for 10 days and stained for ALP. **e** Quantitative measurement of ALP at day 7 and AR at day 21 in mouse-derived cell cultures. Columns represented individual values in each group and the error bars represented standard deviations. $N = 3$ biologically independent samples in control-treated groups and $n = 5$ biologically independent samples in YLL3 and YLL8-treated groups. *$p < 0.05$ between the treated groups versus control (Con) using one-way ANOVA followed by Dunnett's multiple comparison post hoc test. **f** Representative images of ALP staining in mouse BMSCs cultured with beads displaying YLL3 or YLL8 at day 10. Scale bar = 100 µm. **g** Representative images of AR staining in mouse BMSCs cultured with beads displaying YLL3 or YLL8 at day 21. Scale bar = 100 µm. **h** Mouse BMSCs were cultured with YLL3 and YLL8 at $6 \times 10^{-8}$ M in osteogenic media for 10 or 21 days to measure ALP levels at day 10, and AR levels at day 21. Columns represented individual values in each group and the error bars represented standard deviations. $N = 5$ biologically independent samples in control-treated groups, and $n = 6$ biologically independent samples in YLL3 and YLL8-treated groups. *$p < 0.01$ between the treated groups versus control (Con) using one-way ANOVA followed by Dunnett's multiple comparison post hoc test.

**OPC-targeting peptides are anabolic in vivo.** To study whether YLLs would affect bone metabolism in vivo, we injected YLL3 and YLL8 to 4-month-old female and male mice at 10 µg/kg subcutaneously (s.c.), 5x/week, for 21 days ($n = 5–7$/group). hPTH (1–34) was injected s.c. at 25 µg/kg, 5x/week as a positive control. We found the daily injections of YLL3 or YLL8 injection for 21 days did not change body weights (Supplementary Fig. 3c) or cause any visible side-effects. Compared to the PBS-treated mice, both YLL3 and YLL8 increased femoral trabecular bone area measured by bone histomorphometry, in both the females and the males (Fig. 4a). Mineral apposition rate, a parameter corresponding to osteoblast activity, was increased by YLL8 more than 70% in the females and by about 30% in the males, resulting in an overall increase in the surface-based bone formation rate (Fig. 4a, b). The anabolic effect of YLLs was confirmed by increased

trabecular bone mass measured at the distal femur metaphysis by microCT and by serum osteocalcin (Supplementary Fig. 4a). Bone resorption measured by serum CTX-1 suggested that PTH increased bone resorption by 200% compared to the PBS-treated group, especially in the male mice, while YLLs did not change CTX-1 significantly (Supplementary Fig. 4a). Daily YLL3 and YLL8 treatment for 21 days increased the cortical bone mass in the males and increased work-to-failure, in both females and males (Fig. 4d). These results suggested daily injections of low-dose YLL3 and YLL8 induced uncoupling of bone formation and resorption, which increased bone mass in normal adult mice.

**OPC-targeting peptides prevent gonadal deficiency-induced bone loss.** To study whether YLLs would prevent bone loss following estrogen withdrawal in the females (ovariectomy, OVX) or

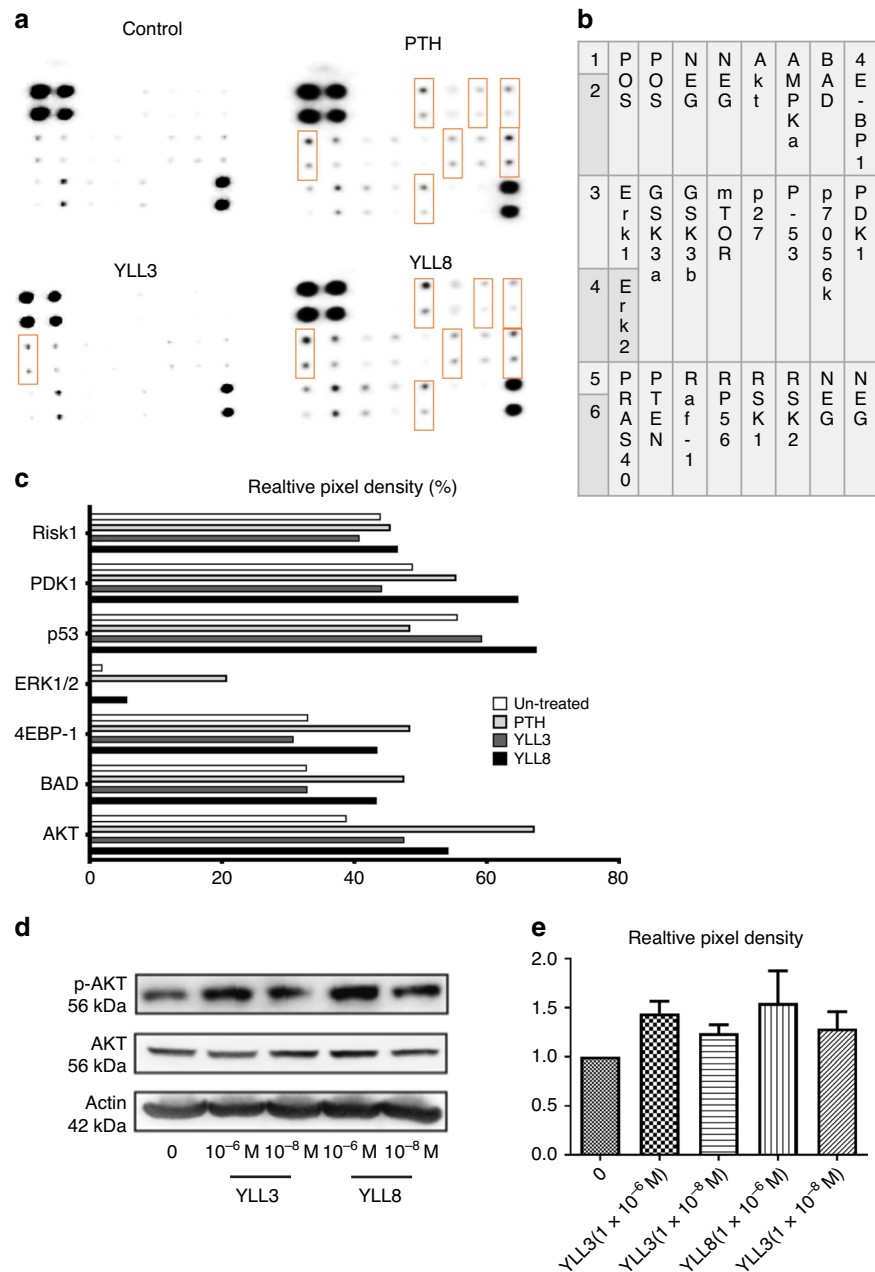

**Fig. 3 YLLs activated the Akt signaling. a** Mouse-derived MSCs were cultured with YLL3 or YLL8 peptides or hPTH (1–34) at $6 \times 10^{-8}$ M in osteogenic medium for 2 h. The cell lysates were collected and used in an Akt protein array, measuring 18 phosphorylated proteins within the Akt signaling pathway. **b** The corresponding protein mapping of the Akt array listed in **a**. **c** Quantitative pixel density/blank control from selected dots highlighted in rectangles in **b**, which were differentially expressed compared to the control. **d** Human-derived MSCs were cultured with YLL3 or YLL8 at indicated concentration in osteogenic media for 2 h. Cell lysates were collected for western blotting of p-Akt, Akt, and beta-actin. Representative gel images from each treatment and **e** quantitative pixel density for p-Akt measured from $n = 3$ independent experiments/group presented in "**d**".

the lack of testosterone in the males (orchidectomy, ORX), we performed OVX or ORX in 8-week-old C57BL/6 mice. Following the surgeries, we injected YLL3 or YLL8 to the OVX or ORX mice at 10 µg/kg, s.c., 5x/week, for 28 days ($n = 6$/group). hPTH (1–34) was injected s.c. at 40 µg/kg, 5x/week as anabolic treatment control. We found the daily injections of both YLL3 or YLL8 for 28 days did not change body weights compared to the OVX/ORX PBS-treated mice (Supplementary Fig. 3d, e) or cause major organ toxicity (data on file). Compared to the OVX PBS-treated mice, YLL3, YLL8, and PTH significantly increased proximal tibial trabecular bone volume by greater than 40%, 30%, and 40% in the OVX mice, respectively (Fig. 5a, b). Daily YLL3 treatment

for 28 days prevented the losses in the trabecular thickness, which was comparable to daily PTH injections (Fig. 5a, b). No treatment significantly changed the mineralizing surface. Mineral apposition rate was increased by >26%, 32%, and 60%, respectively, in YLL3, YLL8, and PTH-treated OVX mice, resulting in overall increases in surface-based bone formation rate (Fig. 5c, d). Similar anabolic effects were observed in ORX mice, where daily injection of YLL3 significantly increased mineral apposition rate and bone formation rate (Fig. 5e). The anabolic effect of YLLs was confirmed by increased trabecular bone area measured at the proximal tibial metaphysis by bone histomorphometry (Supplementary Fig. 4b, c). Circulating bone markers did not change

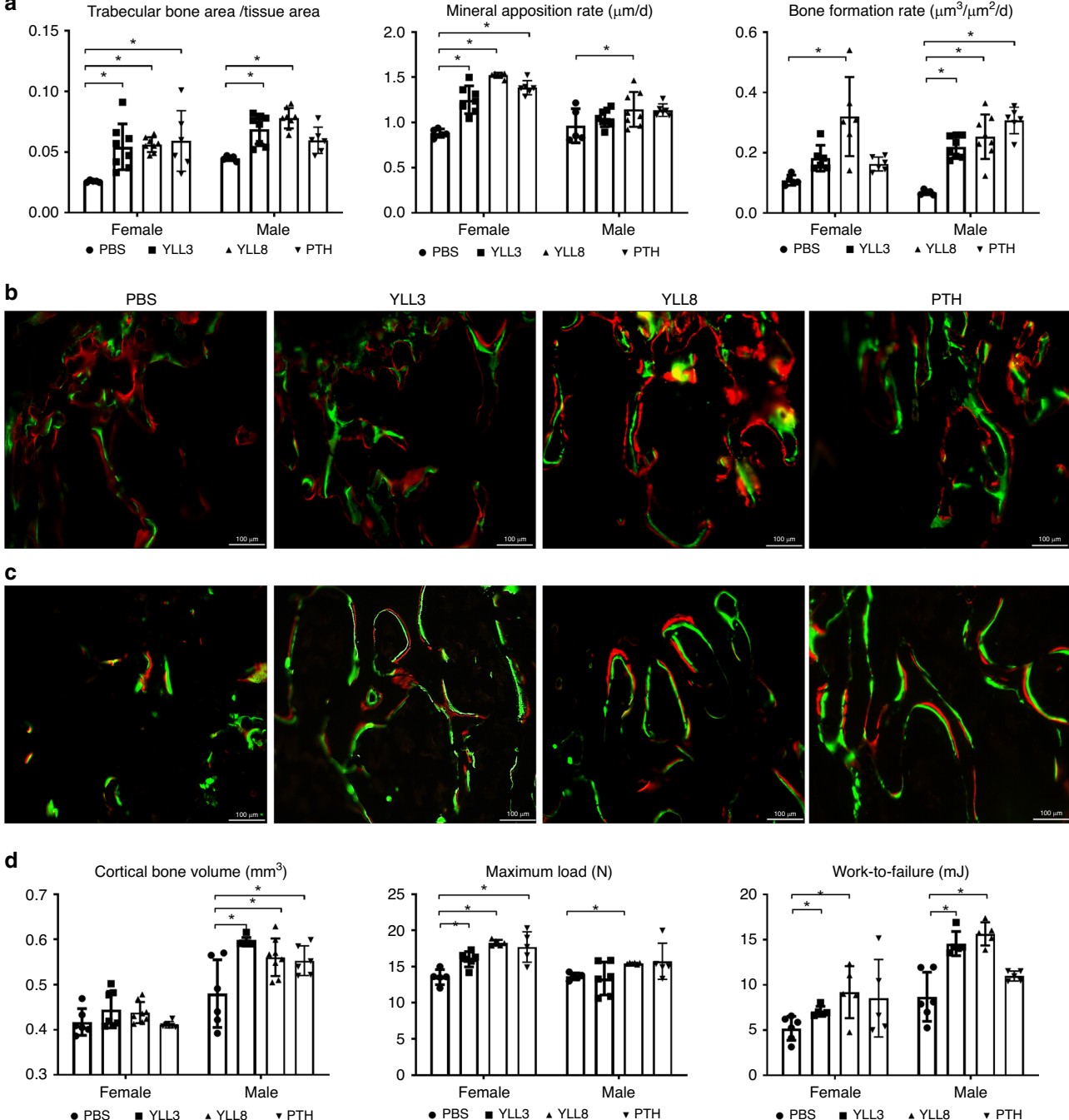

**Fig. 4 YLLs augmented bone formation in vivo in normal adult mice.** Four-month-old female and male mice were treated with PBS control, YLL3 or YLL8 at 10 μg/kg or hPTH(1–34) at 25 μg/kg, s.c., 5x/week for 21 days. All mice received Calcein (green) and Alizarin red (red) at −8 and −1 days before euthanization. **a** Trabecular bone volume and surface-based bone formation were measured in the distal femur metaphysis (DFM). Columns represented individual values in each group and the error bars represented standard deviations. $N = 7$ animals in PBS and PTH-treated groups, and $n = 8$ mice in YLL3 and YLL8-treated groups. *$p < 0.001$ between the treated groups versus PBS-treated group using one-way ANOVA followed by Dunnett's multiple comparison post hoc test for the trabecular bone area/tissue area and bone formation rate; and $p < 0.05$ between the YLL8-treated group versus PBS-treated group for the male Mineral apposition rate using one-way ANOVA followed by Dunnett's multiple comparison post hoc test in the males. **b** Representative images were taken from the DFM trabecular region in the female mice. Scale bar = 100 μm. **c** Representative images were taken from the DFM trabecular region in the male mice. Scale bar = 100 μm. **d** Cortical bone mass was measured at the mid-femurs by microCT; bone strength was obtained by three-point bending of the femurs. Columns represented individual values in each group and the error bars represented standard deviations. $N = 6$ animals/groups. *$p < 0.001$ between the treated groups versus PBS-treated group for the cortical bone volume in the males, maximum load in the females and work-to-failure in the males using one-way ANOVA followed by Dunnett's multiple comparison post hoc test; and $p < 0.05$ between the YLL8-treated group versus PBS-treated group for the cortical bone volume in the males, maximum load in the females and work-to-failure for the maximum load in the males and work-to-failure in the females using one-way ANOVA followed by Dunnett's multiple comparison post hoc test.

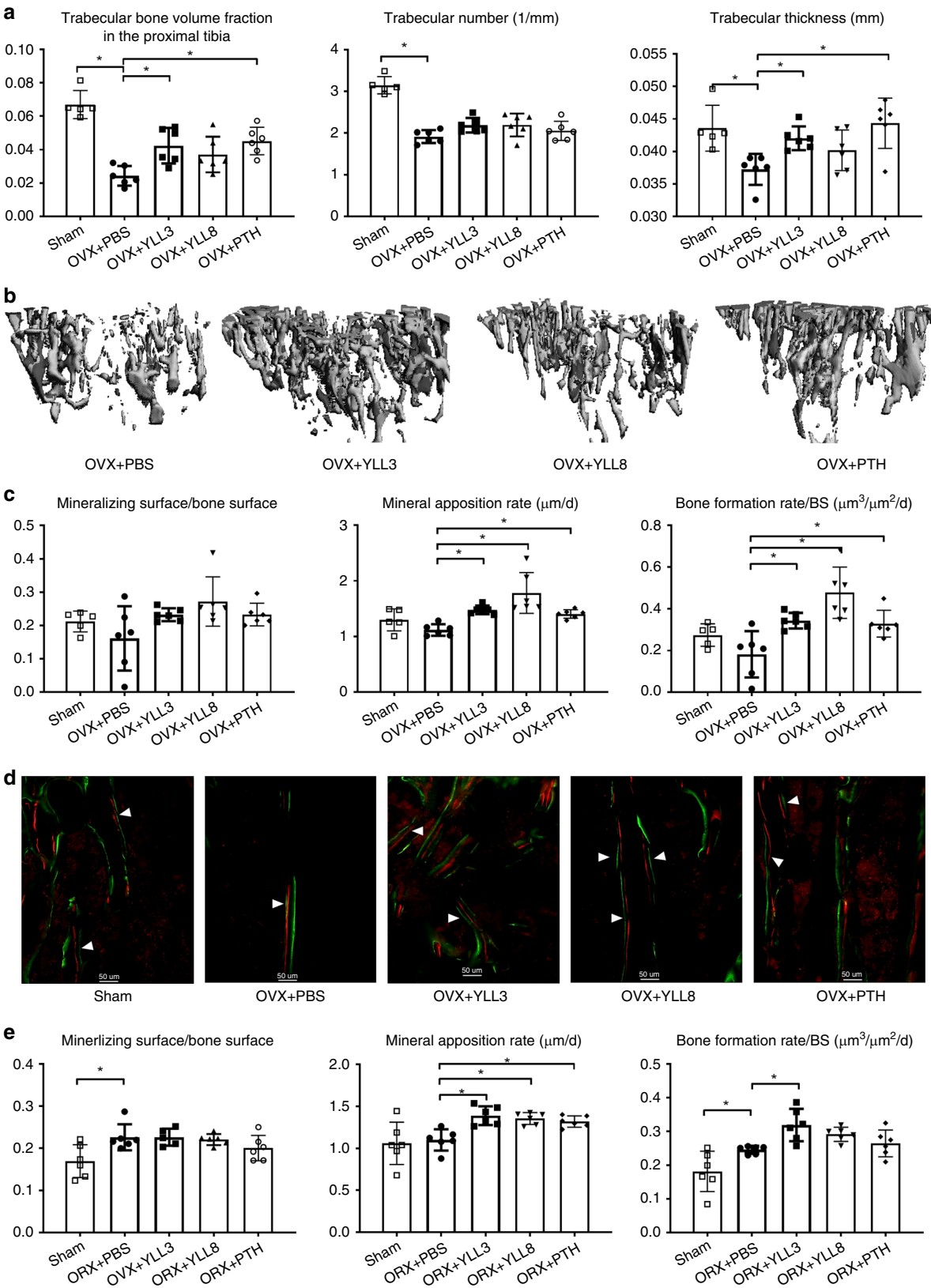

significantly among the groups (Supplementary Fig. 4). These results suggested daily injections of YLL3 prevented bone loss induced by gonadal deficiency in both the female and male mice. Both YLL3 and YLL8 increased bone formation, similar to daily PTH injections.

**OPC-targeting peptides increase callus mineralization.** Since the recruitment and activation of the endogenous osteoblast lineage cells during fracture healing is a crucial step to repair the fracture[29], we next thought to determine if the osteoprogenitor cell-targeting peptides would expedite fracture repair through

**Fig. 5 YLLs prevented bone loss in gonadal-deficient mice.** Two-month-old C57BL/6 mice were ovariectomized (female, OVX) or orchiectomized (male, ORX) and treated with PBS, YLL3, or YLL8 at 10 µg/kg or hPTH (1–34) at 40 µg/kg, s.c., 5x/week for 28 days. All mice received Calcein and Alizarin red at −8 and −1 days before euthanization. **a** Quantitative measurements of trabecular bone microstructure at the proximal tibial metaphysis (PTM). Columns represented individual values in each group and the error bars represented standard deviations. N = 6 animals/group. *p < 0.01 between the treated groups versus PBS-treated group using one-way ANOVA followed by Dunnett's multiple comparison post hoc test. **b** Representative 3D images of the PTM from the female mice in the OVX study. **c** Trabecular bone surface-based bone formation was measured in the PTM in the OVX mice. Columns represented individual values in each group and the error bars represented standard deviations. N = 5 animals in Sham-operated and n = 6 animals/group in the OVXed groups. *p < 0.05 between the treated groups versus PBS-treated group using one-way ANOVA followed by Dunnett's multiple comparison post hoc test. **d** Representative fluorescence images from the PTM trabecular regions in the OVX study. Scale bar = 50 µm. **e** Trabecular bone surface-based bone formation was measured in the lumbar vertebral bodies in the ORX mice. Columns represented individual values in each group and the error bars represented standard deviations. N = 6 animals/group. *p < 0.05 between the treated groups versus PBS-treated group using one-way ANOVA followed by Dunnett's multiple comparison post hoc test.

stimulation of osteogenic differentiation and activities of endogenous osteoprogenitor cells. We used osterix as a marker for osteoprogenitor cells. Since no inducible osterix-reporting mouse is commercially available, we decided to use the inducible Prx1-Cre$^{ERT}$-GFP mouse to track the recruitment and osteogenic differentiation of the osteoprogenitor cells following fracture and YLL3 or YLL8 treatment. Prx1 is expressed by osteoprogenitor cells at the growth plate as well as alongside the trabeculae and endocortical bone surfaces in the un-injured mice[23,30,31], similar to where the osterix is expressed in bone (Supplementary Fig. 5)[27,28]. A closed, stabilized middle-femur fracture model was used to track the activation and osteogenic differentiation of the osteoprogenitor cells that contribute to the bone formation during the healing process[15,32]. Human PTH (1–34) was used as a positive control (40 µg/kg, 5x/week). At day 10 following fracture, we found the presence of Prx1/GFP-expressing cells and were located within the callus, suggesting recruitment and activation of the endogenous osteogenic cells (Fig. 6a). The numbers of Prx1+ cells were greatly expanded, especially in YLL3-treated mice (Fig. 6b), resulting in 100% higher callus bone volume than the PBS-treated mice at day 10 (Fig. 6b).

Further dynamic histomorphometry studies using fluorochrome labeling for mineralization at day 21 post fracture indicated that significantly higher Prx1+ and their descendant cells co-localized with alizarin red mineral apposition, especially in YLL3-treated mice, and co-localized within regenerated bone-callus regions (Fig. 6c). Quantitative measurements by microCT showed higher callus mineral content in YLL3 and YLL8-treated mice at day 21 post fracture (Fig. 6d). These data suggested that YLL8 and PTH had similar effects on activating the osteoblastic lineage cells during fracture healing. In contrast, YLL3 injections substantially activated the osteogenic differentiation of the osteoprogenitor cells and induced significantly higher callus mineralization that expedited the fracture repair.

**The effects of dual bone- and OPC-targeting peptides in vivo.** To further ensure our peptides specifically target bone, we conjugated the YLLs to alendronate (Aln), a bone-affiliate agent, to increase drug delivery to the bone (Supplementary Fig. 6). This method was previously used to bone-targeted delivery of MSC and to increase the exogenous MSC bone homing and engraftment[15–17], or for bone-targeted delivery of drugs[33,34]. The peptide-Aln conjugates using this approach home exclusively to the bone by 6 h following intravenous injection (i.v.)[15]. We next evaluated the effects of the Aln-conjugated osteogenic peptides on bone formation; we treated 2-month-old female mice with PBS control, YLLs-Aln at 300 µg/kg, subcutaneously (s.c.), once every other week x 1 month. YLL3 or YLL8 were used as a daily injection at a dose of 10 µg/kg for 1 month. YLLs-Aln injections did not change body weights or induce any visible side-effects. YLL3, YLL8, and YLL8-Aln increased trabecular bone volume

(p < 0.05 vs. PBS) (Fig. 7a, b). YLL8-Aln increased cortical bone volume and thickness (Fig. 7b, c). YLL3-Aln, on the other hand, lost its anabolic effect when YLL3 was conjugated to the alendronate. Taken together, these results suggest a bisphosphonate-conjugation of YLL8 peptide decreases the dosing frequency but maintained similar or better osteogenic effects on bone compared to daily injections of the peptide itself.

## Discussion

Here, we use a dual affinity and functional screening method to discover "osteogenic-specific" peptides[14]. Two of these leading osteogenic peptides, YLL3 and YLL8, not only had a high affinity for osterix+ cells but also activated phosphorylation of Akt, a pro-survival signal for the osteogenic progenitors. Both YLL3 and YLL8 increased differentiation and maturation of the osteoblasts in vitro. Short-term (3 weeks) daily injections of low-dose YLL3 and YLL8 induced comparable bone anabolic action to PTH (1–34): increased key bone formation indices, mineral apposition rate, corresponding to osteoblast activity, and bone formation rate. As opposed to PTH (1–34), the daily injection of YLLs did not significantly affect bone resorption. This uncoupling of bone-remodeling resulted in rapid modeling-depending bone gain and increased bone strength in both trabecular and cortical bone, which were higher in YLLs-reated mice compared to that of hPTH (1–34)-treated mice. Daily injections of YLLs for 21–28 days, especially YLL3, prevented bone loss induced by gonadal hormone deficiency and increased callus formation and mineralization during fracture repair. Conjugation of YLL8 to Aln alleviate the need for repeated injection; a single injection of YLL8-Aln was able to achieve a similar degree of bone gain in both trabecular and cortical bone, as daily injections of unconjugated YLL8. Together, these data suggest YLLs can be used as a "therapeutic peptide", or be deployed with a bone-targeting agent to further its bone specificity[15–17].

Current pharmacologic treatment options for osteoporosis include two main categories: anti-resorptive and anabolic treatments. The recombinant human PTH (1–34) (Teriparatide) is best known for its bone anabolic actions and consists of 34 amino acids, which is the bioactive portion of the hormone. The more recently FDA-approved PTHrP (Abaloparatide) is a 139–173-amino acid protein with N-terminal homology to PTH. Both hPTH(1–34) and PTHrp have significant anabolic effects on bone[35–38] but also have other widespread physiological actions and expressed in a diversity of tissues[39,40]. Besides their effects on bone and joint development, PTH and PTHrp act in an autocrine/paracrine fashion to regulate calcium metabolism and organogenesis such as mammary gland development[41]. Sclerostin-antibody, on the other hand, is another promising bone anabolic agent, and its actions may be more specific to bone cells[10,42–45]. Sclerostin is expressed almost exclusively in bone by osteocytes[46,47]. Recently, sclerostin is found to express in the

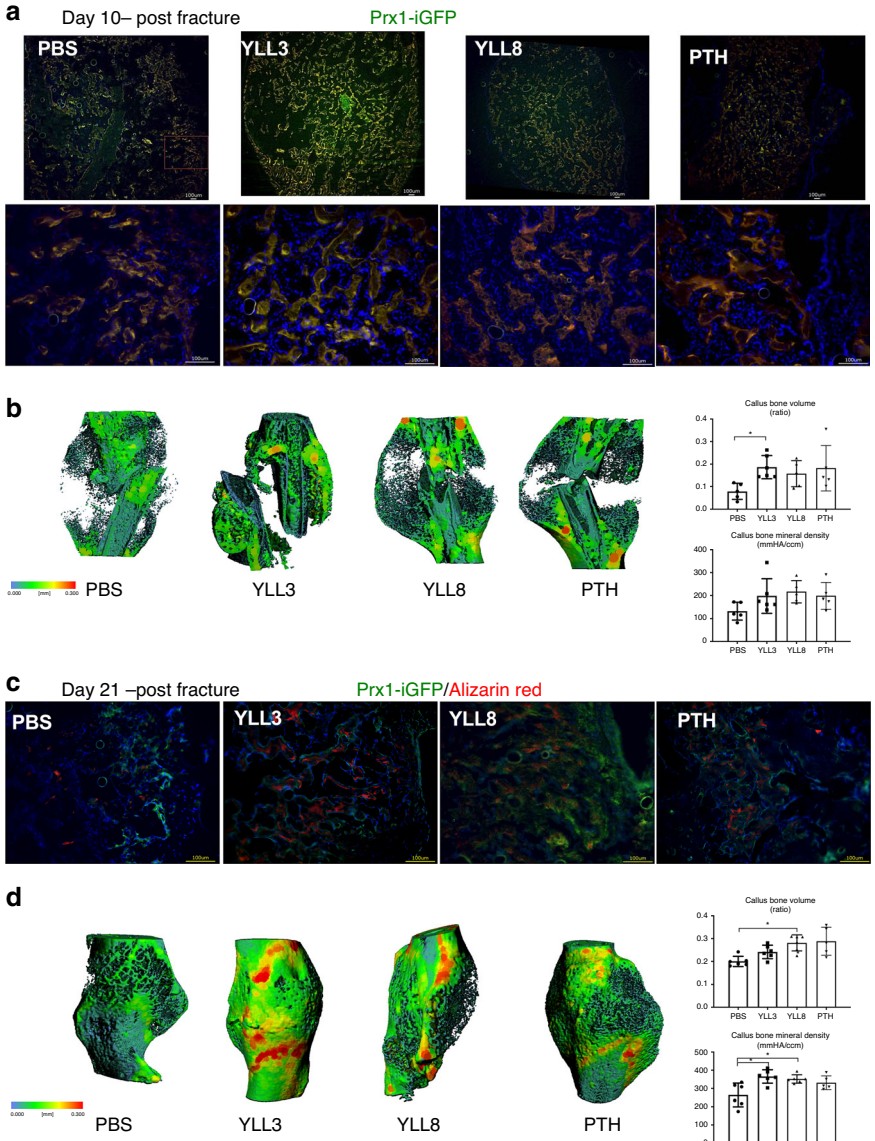

**Fig. 6 YLL3 induced higher callus mineralization during fracture healing.** Female Prx1$^{ERT}$-GFP mice were fractured at 2 months of age and received tamoxifen at 10 mg/kg for 3 days. YLL3 or YLL8 were given at 10 μg/kg, and hPTH (1–34) was given at 40 μg/kg, s.c., 5x/week for 10 or 21 days. **a** Representative callus images from the outer edge of the callus in indicated treatment groups at day 10 post fracture viewing under low power (4x) or high power (20x). Scale bar = 100 μm. **b** Representative microCT thickness mapping of the callus, callus bone volume, and bone mineral content from indicated treatment groups at day 10 post fracture. Columns represented individual values in each group and the error bars represented standard deviations. $N = 5$ animals group. *$p < 0.05$ between the treated groups versus PBS-treated group using one-way ANOVA followed by Dunnett's multiple comparison post hoc test. **c** Representative callus images from the outer edge of the callus in indicated treatment groups at 21 days post fracture. Alizarin red was given at −1 day before euthanization. Scale bar = 100 μm. **d** Representative microCT thickness mapping of the callus, callus bone volume, and bone mineral content from indicated treatment groups at day 21 post fracture. Columns represented individual values in each group and the error bars represented standard deviations. $N = 6$ animals/group in PBS, YLL3, and YLL8-treated group and $n = 5$ in PTH-treated group. *$p < 0.05$ between the treated groups versus PBS-treated group using one-way ANOVA followed by Dunnett's multiple comparison post hoc test.

cartilage[48,49] and lymphocytes[50,51], which might raise concerns on sclerostin-antibody's extra-skeletal side-effects.

The canonical Wnt/β-catenin signaling pathway plays a critical effect on bone formation and is an active drug target for researchers searching for bone anabolic therapeutic drugs. We chose to focus on Akt activation because it was one of the lead kinases activated by α4β1 integrin upon binding to osteoprogenitor cells[15–17]. Also, we were more interested in activating the osteoprogenitor cells and their osteogenic potential than to induce cell mitogen or to induce de novo bone formation, which was observed using Wnt-targeting or growth factors[32,52–55]. To verify the effect of our "osteogenic peptides", YLLs, on osteoblast differentiation, we performed ALP staining and AR staining, which reflect the initial and terminal phases of osteoblast differentiation, respectively. We also confirmed the activation of the Akt signaling pathway, which is necessary for bone anabolic effects[17,56,57]. YLL8 has broad effects on Akt pathway activation, which was very similar to PTH, while YLL3 is more specific for Akt activation. Both YLL3 and YLL8 increased osteogenic differentiation of the osteoprogenitor cells in vitro. We further demonstrated that injections of YLL3 and YLL8, increased mineral apposition rate, a parameter reflecting active osteoblast

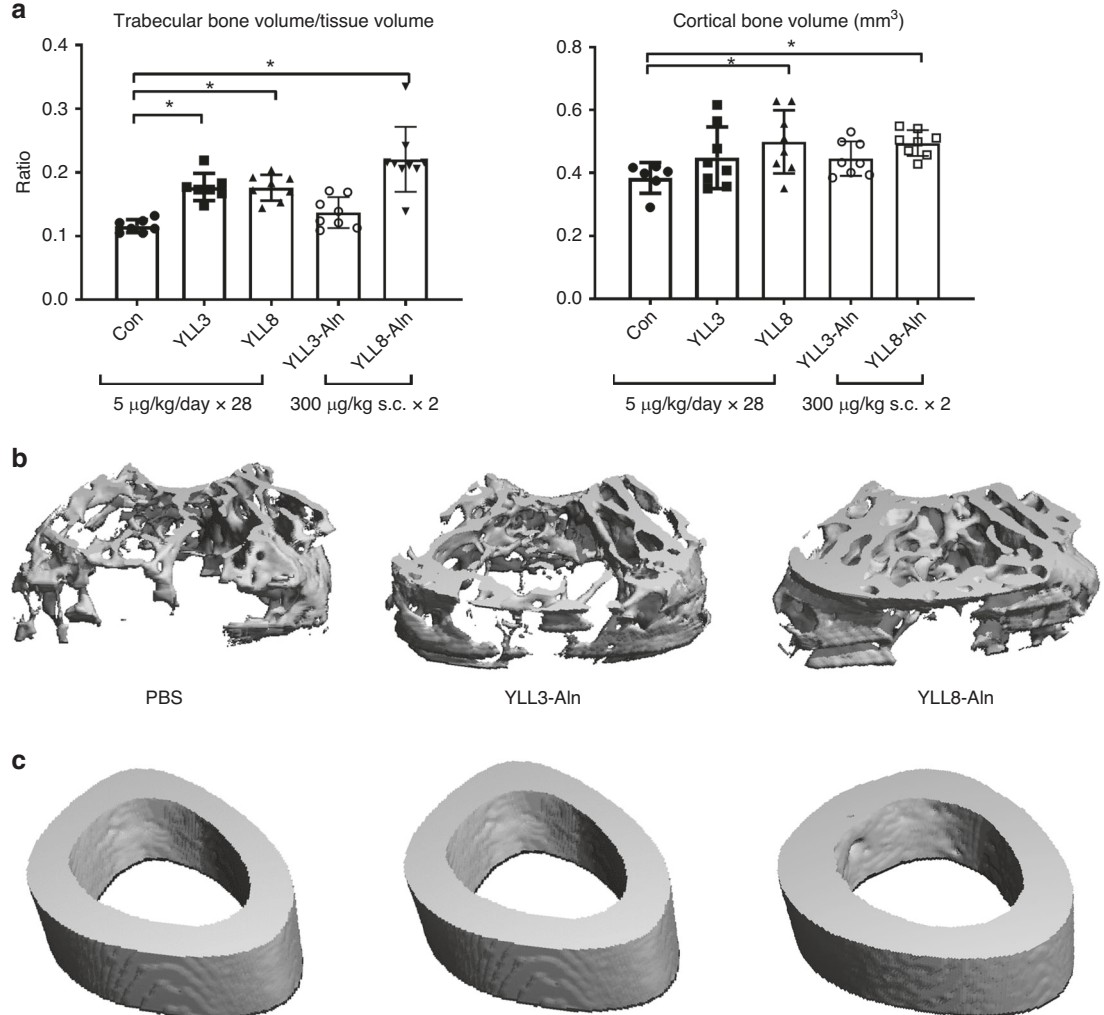

**Fig. 7 YLL8-Aln increased trabecular and cortical bone mass.** Two-month-old female mice were treated with PBS control, YLL3, or YLL8 at 10 μg/kg or YLL3-Aln or YLL8-Aln at 300 μg/kg s.c., every other week for 1 month. Mice were euthanized at day 28. **a** Quantitative distal femoral trabecular and middle-femur cortical bone volume measured by microCT. Columns represented individual values in each group and the error bars represented standard deviations. $N = 8$ animals/group. *$p < 0.05$ between the treated groups versus PBS-treated group using one-way ANOVA followed by Dunnett's multiple comparison post hoc test. **b** Representative 3D images of the trabecular region taken from the distal femurs in the indicated treatment groups. **c** Representative images of the middle femoral cortex in indicated treatment groups.

function, and resulted in increased bone formation, bone mass, and bone strength in young mice. Our relatively short-term treatment (21 days) showed the YLLs had higher anabolic effects on the male mice than the female mice in general, which could be due to young males usually have greater osteogenesis than the females and that high estrogen levels during growth may suppress bone formation[58,59].

We conjugated the YLLs to a bone-targeting drug, alendronate (Aln). In this case, Aln was used as a "carrier" for bone-specific delivery of the peptide. The bisphosphonates (BPs) are known to contain two phosphonate groups with covalently bonded side chains known as $R_1$ and $R_2$. The presence of a hydroxyl (OH) in the $R_1$ chain gives an affinity to the bone. The anti-resorptive potency of BPs is believed to reside in the $R_2$ chain, especially for the N-containing BPs[60]. In our peptide-BP conjugation, we used the $NH_2$ in the $R_2$ chain to link Aln to the peptide. The published studies using similar methods also did not observe anti-resorptive effects from these peptide-BP conjugation[15–17,61]. This Aln-conjugation was deposited to the skeleton within 6 hours. After 7 days, the drug remained primarily in the skeleton[16]. To further determine the enhancing bone-targeted effect of YLLs, we

performed an in vivo experiment and demonstrated two s.c. doses of YLL8-Aln increased both trabecular and cortical bone mass. Collectively, these studies demonstrated that YLL8 could stimulate osteoblast differentiation in vitro and in vivo, demonstrating the efficacy of YLL8-Aln on bone formation. Dynamic bone histomorphometric and histologic analyses confirmed that YLL8 significantly promoted bone formation via enhancing the osteoblast activity. Surprisingly, YLL3-Aln lost its anabolic effects on bone when YLL3 was conjugated to Aln, the conjugation probably changed the chemical configuration of YLL3 necessary for bone formation, or higher dose/increased dosing frequency may be needed to achieve the anabolic effects observed for the peptide itself. Further dose-defining studies may require to better define the potential anabolic action for YLL3-Aln. Alternatively, we can modify our conjugation method so that the YLL3 can be cleavable from the Aln conjugation such that the free peptide could be released after it reaches the bone.

In the United States alone, nearly 8 million adults over year experience fracture, of which ~5–13% result in nonunion or delayed union[62]. Current therapies for the treatment of fracture non-unions consist mainly of bone grafts and the use of

recombinant bone morphogenetic proteins (BMPs). Bone grafts have significant complication rates ranging from 10–25% and are associated with inconsistent outcomes. BMP treatment requires orthopedic operation before BMP delivery. BMPs can cost inflammation, induce poorly mineralized, and is costly[63,64]. There is a clear need for bone-building anabolic agents that can increase bone mineralization to expedite fracture repair. To this end, we found the daily injection of YLL3 stimulated more mature callus formation at an early stage (day 10 post fracture) and higher callus mineral density at a later stage (day 21 post fracture). Since the initial phase of fracture repair is the recruitment and activation of the osteoprogenitor cells, we found mice treated with YLL3 increased the number of osteoprogenitor cells in the callus at the early stage, suggesting an activation of these endogenous osteogenic cells that were recruited to the fracture sites. The YLL3-treated mice formed well-mineralized lamellar tissue with higher mineral apposition at day 21 compared to the more apparent woven bone formation in PBS-treated controls. Fractures unionized at day 21 post fracture in YLL3-treated mice, which was not observed until after 35–45 days in PBS-treated mice[15,65,66], and was superior to daily injections of PTH used at 40 μg/kg. Unfortunately, our fracture study was not powered to identify changes in the biomechanical properties of the fracture callus. However, since bone mineral density, but not the callus size, was the main determinant for bone strength[15], we anticipated a higher bone strength would result from higher bone mineral density following YLL3 treatment.

In conclusion, we have developed a focused OBOC library screening method to identify osteoprogenitor cell-specific targeting osteogenic peptide through a pro-cell surviving mechanism. The bone-seeking function is achieved by linking the bisphosphonate alendronate to YLL8. The same "one-entity-two-action" screening principle can be used to screen antagonist or agonist drugs targeting other signaling pathways. The two leading osteoprogenitor cell-targeting peptides, YLL3 and YLL8, increased osteoblast differentiation and maturation in vitro, and increase osteoblast activities in vivo. We can also further improve bone specificity by conjugation of the peptide to a bisphosphonate that sustains the anabolic effects but lowers the drug dosing frequency. The osteogenic peptide might be a potential bone anabolic agent against to treat osteoporosis or to expedite fracture repair.

## Methods

**Synthesis of focused OBOC library**. The synthetic approach of the focused OBOC library is shown in Supplementary Fig. 1. The OBOC library was synthesized on TentaGel S $NH_2$ beads using standard solid-phase peptide synthesis (SPPS) approach employing split-mix strategy and Fmoc chemistry as previously reported[12]. Four equivalents of Fmoc-amino acids were used for coupling in the presence of 6-chloro-1-hydroxybenzotriazole (6-Cl HOBt) and $N,N'$-diisopropylcarbodiimide (DIC). Fmoc was removed with 20% 4-methylpiperidine in $N,N'$-dimethylformamide (DMF). Side chain protecting groups of amino acids were removed for 2 h with a cocktail containing 82.5% trifluoroacetic acid (TFA): 5% thioanisole: 5% phenol:5% water: 2.5% triisopropylsilane (TIS) (v/v). After neutralization with 2% DIEA/DMF twice, the resin was washed sequentially with DMF, MeOH, dichloromethane (DCM), DMF, 50% DMF/water, water, and ethanol. The bead library was stored in 70% ethanol. The amino acids used in the report were listed in Supplementary Tables S1–4.

**Solid-phase synthesis of YLL3 and YLL8**. YLL8 was synthesized on Rink amide MBHA resin (GL Biochem, Shanghai, China) using a similar approach as previously reported[12]. The compound was cleaved off the beads with a cleavage mixture of 95% TFA: 2.5% water: 2.5% TIS. The cleavage reaction was conducted at room temperature for 2 h. The liquid was collected and concentrated. The crude product was precipitated with cold diethyl ether and purified using preparative reversed-phase high-performance liquid chromatography (RP-HPLC). The fraction was collected and lyophilized to give designed product YLL8, MALDI-TOF MS: 813.30 daltons.

YLL3 was synthesized using a similar approach as YLL8 but coupled the following different building blocks Fmoc-Val-OH, Fmoc-Ser(tBu)-OH, Fmoc-Glu (OtBu)-OH, Fmoc-Cit-OH, and UPA sequentially to beads. The crude YLL3 was cleaved off the beads and purified by RP-HPLC. Matrix-assisted laser desorption/ionization-time of flight mass spectrometry (MALDI-TOF MS): 756.25 daltons.

**Synthesis of YLL3-Aln and YLL8-Aln**. The conjugation of YLL3 and YLL8 with Aln was achieved via maleimide-SH Michael addition, as previously reported[17]. The synthesis involved in three steps: (1) Solid-phase synthesis of YLL8-Lys(D-Cys) or YLL3-Lys(D-Cys); (2) Synthesis of Aln-maleimide; (3) Conjugation of YLL8-Lys (D-Cys) or YLL3-Lys(D-Cys) with Aln-maleimide. MALDI-TOF MS of YLL8-Aln: 1972.75 daltons. MALDI-TOF MS of YLL3-Aln: 1915.76 daltons.

**MSC isolation, expansion, and osteogenic differentiation**. Mouse bone marrow aspirates were obtained from Osx-mCherry reporter mice, which were obtained via an MTA agreement with Dr. Peter Maye at the University of Connecticut Health Center. Bone marrow stromal cells (BMSCs) were maintained in MSC expansion media (R&D Systems USA) for three to five passages before experimentation. The BMSCs were then plated into tissue culture flasks supplemented with YLLs (3–6 x $10^{-8}$ M) for up to 28 days.

Bone marrow aspirates were obtained from healthy human donors (StemExpress, Placerville, CA). The bone marrow mononuclear cells (MNCs) were isolated using a density gradient centrifugation, and then MNCs are directly plated into tissue culture flasks. MSCs were then expanded for at least three to five passages before experimentation. MSCs were then plated into tissue culture flasks supplemented with YLL3 or YLL8 ($1 \times 10^{-9}$ M–$1 \times 10^{-6}$ M) for up to 21 days.

For osteogenesis quantification, alkaline phosphatase (ALP) activity is measured at days 7–10[24]. Matrix mineralization is determined on day 21 using Alizarin Red. The optic density of the supernatants was measured at 405 nm[16,17,23,24].

**MTS experiment**. Cells were plated in 96-well plates ($5 \times 10^3$/well) overnight and then subjected to treatment with increasing doses of YLLs (0, 1, 5, 10, 20 μM) in mesenchymal maintenance medium for three different durations (24, 48, and 72 h). After incubation, MTS working solutions were added in each well and incubated for another 2 h before the plate reading (490 nm, SpectraMax M2, Molecular Devices, USA). Cell viability was calculated relative to the control.

**Binding specificity of YLLs for osteoprogenitor cells**. The affinity of YLLs for osteoprogenitor cells (OPC) was evaluated semi-quantitatively by incubating MSCs derived from mice and humans and were cultured in an osteogenic medium that contained 0.2 mM ascorbic acid and 10 mM β-glycerophosphate for 5 days and then incubated with beads displaying YLL3 or YLL8. Semi-quantitative assessment of binding affinity was performed at 2, 6, and 12 h. Similar semi-quantitative binding affinity was performed for the peripheral mononuclear cells.

**The specificity of Akt signaling activation by YLLs**. We confirmed the activation of Akt signaling by western blot using an Akt signaling array following the manufacturer's instruction (www.raybiotech.com). This kit was a slide-based antibody array founded upon the sandwich immunoassay principle. Each dot represented phosphorylation protein within the Akt signaling pathway that was listed in Fig. 3b. Target-specific capture antibodies, biotinylated protein (positive control), and nonspecific IgG (negative control) were included in duplicate onto the glass slides.

Western blot analyses were performed to measure total and phosphorylated Akt and total Akt proteins. Cells were lysed in cold buffer containing 20 mM Tris (pH 7.5), 150 mM NaCl, 1 mM EDTA, 1 mM EGTA, 1% Triton X-100, 2.5 mM sodium pyrophosphate, 1 mM β-Glycerophosphate, 15 mmol/L NaF, 1 μg/mL Leupeptin, 1 mM PMSF, and centrifuged at $12,000 \times g$ for 10 min. The protein extracts (10 μg/lane) were subjected to electrophoresis on 4–10% sodium dodecyl sulfate–polyacrylamide gel electrophoresis gel and transferred to a polyvinylidene fluoride membrane, which was stained by Naphthol Blue-Black to confirm equal protein loading. The membranes were blocked in Bovine serum albumin for 1 h and then incubated with the following primary antibodies: Akt, phosphor-Akt (Ser473) or beta-actin (Cell Signaling Technology, Beverly, MA, USA). Membranes were then incubated with horseradish peroxidase-conjugated goat anti-rabbit or anti-mouse IgG secondary antibody, and detection was done using peroxide solution (Pierce, Rockford, IL USA). Films were scanned, and band densities were analyzed using a BIO-RAD ChemiDoc$^{TM}$ MP Imaging system (Bio-Rad Life Science Research, Hercules, CA, USA).

**Animal studies**. Effects on intact animals. Four-month-old mice of C57BL/6 background were treated with either PBS (both sex), YLL3 (10 μg/kg, 5x/week, s.c., 5x/week both sex), YLL8 (10 μg/kg. 5x/week, s.c., 5x/week both sex), and human PTH (1–34) (25 μk/kg, 5x/week, s.c., 5x/week both sex). Mice were sacrificed at week 4. Calcein (10mg/kg) and Alizarin Red (20 mg/kg) were given to all the mice s.c. at −8 and −1 days before euthanization.

*Hormonal deficiency-induced bone loss study*: This in vivo experiment was performed in 8-week-old mice of C57BL/6 background. The mice were

ovariectomized or orchiectomized at 8-week of age, and then treated with YLL3 (10 μg/kg, i.p., 5x/week, both sex), YLL8 (10 μg/kg. s.c., 5x/week, both sex), and human PTH (1–34) (40 μk/kg, s.c., 5x/week, both sex). Mice were sacrificed at week 4. Calcein (10 mg/kg) and Alizarin Red (20 mg/kg) were given to all the mice s.c. at −8 and −1 days before euthanization.

*Fracture study*: Prx1-Cre^{ERT}-GFP mouse was obtained from the Jackson Laboratory (Jackson 029211). Closed transverse diaphyseal fractures of the right femur were generated in 8-week-old mice using a previously described method with some modification[67]. Briefly, a 0.38 mm-diameter stainless-steel pin was inserted into the medullary canal. Fractures were created at the mid-femur using a drop-weight blunt guillotine device[67,68]. The success of the fractures and pin placement was confirmed by X-ray (Faxitron, Lincolnshire, IL, USA). Female mice were fractured and treated with placebo (phosphate-buffered saline, PBS), YLL3 (10 μg/kg, s.c., 5x/week), YLL8 (10 μg/kg, s.c., 5x/week) or hPTH (1–34) (40 μg/kg, s.c., 5x/week). All mice received Tamoxifen at 10mg/kg at days −1 to 1 during fracture. Mice were euthanized on days 10 and 21 post fracture. Alizarin Red (20 mg/kg) was given to all the mice s.c. at −8 days before euthanization.

*YLL-Aln experiment in vivo*: 8-week-old female mice of 129SvJ background (osterix-mCherry wild type mice) were treated with either PBS, YLL3 (10 μg/kg, s.c., 5x/week), YLL8 (10 μg/kg, s.c., 5x/week, YLL8-Aln (300 μg/kg, s.c. every other week x two times), or human PTH (1–34) (40 μk/kg, s.c., 5x/week). Mice were sacrificed at week 4 (n = 6–10/group). Alizarin Red (20 mg/kg) and calcein (10mg/kg) were given to all the mice s.c. at −8 and −1 days before euthanization.

The Institutional Animal Care and Use Committee at UC Davis approved all animal procedures.

**Micro CT measurement for bone architectures**. The right distal femurs and middle femurs were scanned by microCT (VivaCT 40, Scanco Medical AG, Bassersdorf, Switzerland) at 70 KeV and 145 μA at an isotropic resolution of 10.5 μm in all three dimensions, with an integration time of 250 ms. The outer boundary of the trabecular bone within secondary spongiosa was manually defined by contouring. Gaussian filtering with Sigma 0.8 and support one was used to minimize the image noise. We used different thresholds to define trabecular (250) and cortical bone (350). The same settings and thresholds were used for all samples[16,17,69]. Total tissue volume (TV), bone volume (BV), bone volume fraction (BV/TV), and trabecular (Tb.Th) or cortical bone volume were reported.

We used 50 KeV and 145 μA at an isotropic resolution of 10.5 μm in all three dimensions, with an integration time of 300 ms for the fracture study. The entire callus was scanned centered at the fracture line. Gaussian filtering with Sigma 0.8 and support one was used to minimize the image noise. We used thresholds ranged between 150 and 230 to define the newly formed callus at day 10, and the threshold range of 230–450 to define the newly mineralized callus at day 21. Total tissue volume (TV), callus volume (CV), callus volume fraction (CV/TV), and callus mineral density were recorded[67,68].

**Bone histomorphometry**. The right distal femurs were collected, fixed in 4% formaldehyde with 10% sucrose (W/V) at 4 °C for 2 days, further dehydrated in 30% sucrose overnight and embedded in Optimal Cutting Temperature (O.C.T.) medium (Thermo Fisher, Waltham, MA, USA). Cryosections were prepared and imaged with a Keyence BZ-X9000 all-in-one Fluorescence Microscope (Itasca, IL, USA). Surface-based bone formation was quantitated using Bioquant (Bioquant Image Analysis Corporation, Nashville, TN, USA) following the ASBMR bone histomorphometry guideline[70].

**Serum bone marker measurements**. Serum levels of osteocalcin and CTX-1 were measured by ELISA (Biomedical Technologies, Stoughton, MA, USA). All measurements were performed in duplicate for each sample following the manufacturer's instructions.

**Near-infrared fluorescence imaging studies**. YLL8-Aln–DiD direct-conjugated compound or YLL8-DiD controls were injected via the tail vein to the mouse. Animals were euthanized, and the high limbs were dissected and placed on a sheet of transparency. Images were acquired with a Kodak IS2000MM Image station (Rochester, NY, USA) with excitation filter 625/20 bandpass, emission filter 700WA/35 bandpass, and 150 W quartz halogen lamp light source set to maximum. Images were captured with a CCD camera set at F stop = 0, FOV = 150, and FP = 0.

**Statistical methods**. All data analyses were performed using Prism GraphPad, version 8.0 (San Diego, CA USA). All inclusion/exclusion criteria were pre-established, and no animals or samples were excluded from the analysis. We used our previous data on mice who underwent ovariectomy and received PTH treatment to calculate sample size with power set at 0.8, a = 0.05 for in vivo studies[16]. Data were presented as mean ± standard deviation (SD). Two -way ANOVA with mixed-effects analysis was used for the parameters presented in Fig. 4 to test the difference with the control-treated group using Dunnett's multiple comparison test within the same sex. For the rest of the data, one-way ANOVA was used to compare the population mean of the outcome variable of interest among all groups. If the overall test was statistically significant, made pairwise comparisons were

made comparing to the control-treated group using Dunnett's multiple comparison post hoc test[15].

**Reporting summary**. Further information on statistics, software and code, data reporting, study designs, and materials is available in the Nature Research Reporting Summary linked to this article.

## Data availability
Source data are provided with this paper.

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

## Acknowledgements

This work was funded by NIH grant R01 AR061366 (to W.Y.) and bridge fund provided by UC Davis School of Medicine for R01 AR061366 (to W.Y.).

## Author contributions

M.J.: collection and assembly of data, data analysis and interpretation, and final approval of the manuscript. R.L.: library synthesis and screening, data analysis and interpretation, and final approval of the manuscript. X.P.L.: collection and assembly of data, data analysis and interpretation, and final approval of the manuscript. A.K.: collection and assembly of data, data analysis and interpretation, manuscript writing, and final approval of the manuscript. W.W.X.: collection and assembly of data, data analysis and interpretation, manuscript writing, and final approval of the manuscript. L.X.L.: data interpretation, and final approval of the manuscript. Y.P.L.: collection and assembly of data, data analysis and interpretation, and final approval of the manuscript. K.L.: conception and design, data analysis and interpretation, and final approval of the manuscript. W.Y.: conception and design, collection and assembly of data, data analysis and interpretation, manuscript writing, and final approval of the manuscript.

## Competing interests

The authors declare no competing interests.
