## [Peer Review File · Nature Communications]

Reviewers' Comments:

Reviewer #1:

Remarks to the Author:

This paper reports the result of a novel screen designed to provide peptides that both bind to the surface of bone marrow stromal cells (BMSCs) and also activate the Akt pathway in doing so. The authors seem to indicate the binding is expected to be through integrin $\alpha4\beta1$, since the combinatorial peptide library is biased around "LDV" and "QIDS" motifs, that are apparently known to be present in $\alpha4\beta1$ -binding peptides. The best peptide hits are then shown to stimulate some aspects of bone cell metabolism and development in vitro and in vivo.

The screen described is a significant extension of a platform described by the Lam lab previously for scoring not only cell binding to a bead-displayed ligand, but demanding a functional output of that binding event as well. Previously, a rather crude experiment was published in which peptides that induce cell death in a cancer line upon binding the bead were identified. This study, in which a particular signaling pathway is activated, is much more appealing. Also, the peptides isolated in the original study were not at all potent, but the ones described here appear to have reasonable activity (but see point 5 below).

Overall, this paper is a novel contribution to the development of new screening technology and appears to have provided some interesting lead peptides with good activity in promoting bone formation.

The paper does have some weaknesses. It is not very well written. Sections are difficult to follow and the introduction seems to have a lot of extraneous information. There are many gaps in explaining what was done and why (see below), including why activation of the akt pathway was set as a goal in the screen. True, if one were an expert in the topic, some of this would be understood, but a general reader will quickly be lost.

1. Please provide an indication of the composition of the peptide library in the main text. There is a non-peptidic, urea-containing moiety on the N-terminal end. How did this come about? What was the size of the library and what amino acids were employed in its construction?
2. Please provide references for the LDV and QIDS preference of the $\alpha4\beta1$ receptor.
3. Please show a simple picture (dark field image) of the bead-cell complex that corresponds to the fluorescence micrograph shown in Fig. 1a.
4. The text refers to a Fig. 2e, but there is no panel e in Fig. 2.
5. It would be good to see a dose-response of the induction of the Akt pathway by the selected peptides. From the data presented in the paper, it is really not clear how potent the peptide is.
6. Fig. 3 is difficult to follow. It would be so much better just to provide the grid on Fig. 3b. rather than having Fig. 3a be a separate legend of sorts. Also, please explain the significance of dots being there or not. There are red boxes in Fig. 3B, but their significance is not explained.

In general this is an interesting study, but the paper should be redone extensively to improve its readability and fill in gaps in explaining to the reader exactly why things were done.

Reviewer #2:

Remarks to the Author:

This study developed a screening method for identifying osteoprogenitor-specific peptides. The authors developed it as an "one-entity tri-action" screening principle. Two targeting peptides were screened, and defined as YLL3 and YLL8. In vitro and in vivo study showed that they could increase osteogenic differentiation and osteoblastic activity. They also conjugated the two targeting peptides with bisphosphonate, which sustained bone anabolic action but lower drug dosing frequency. The work was original and show potential for translational application. However,

the following concerns limited the novelty of this study and several critical evidences were missing in this manuscript.

1. Definition of osteoprogenitor was not clear. In Figure 1, MSCs were induced for three days and then used for screening. At day 3, how many MSCs were transformed to osteoprogenitors was not clear. The authors used osterix as a marker for osteoprogenitors and showed good binding affinity of the screened peptides with osteoprogenitors rather than lymphocytes. Whether the screened peptides could recognize MSCs need to be confirmed.
2. Evidences from this study showed that the screened peptides activated AKT signaling in osteoprogenitors. What is the mechanism for the agonistic action of the screened peptides? Is the mechanism same for the screened YLL3 and YLL8?
3. In Figure 5, in vivo data showed that the two peptides (YLL3 and YLL8) could increase mineral apposition rate, bone formation, bone mass, and bone strength in young mice as well as expedited fracture repair. Could they directly recognize bone and osteoprogenitors with conjugating Aln? What is the rationale for conjugating YLL3 and YLL8 with Aln?
4. In vivo biophotonic imaging and Immunofluorescent staining data need to be provided to show the bone and osteoprogenitors targeting.
5. The authors suggested that YLL3 and YLL8 were pro-survival ligands which stimulated osteogenesis via enhancing the AKT signaling, it is important to provide evidences that YLL3 and YLL8 could enhance proliferation and survival of the osteoprogenitors, which were missing in this manuscript.
6. In Figure 6, YLL3 and YLL8 could be chosen as controls to show whether the conjugation of Aln compromise the anabolic action of YLL3 and YLL8.
7. For the animal models, the authors chose young mice and bone fracture mice, it is critical to examine the bone anabolic action in aged osteoporotic mice or OVX mouse models.
8. The authors used mouse MSCs when screening peptides. Binding affinity of the screened peptides with human or rat cells should be tested to confirm the translational potential of the peptides.
9. In Figure 1, the authors demonstrated that they identified 22 peptides and focused on two leading peptides (YLL3 and YLL8, Fig. 1d) based on their high binding affinity to osterix cells as well as their osteogenic effects in vitro. However, evidences of their osteogenic effects were missing.
10. Figure 2d was missing.
11. In Figure 4, the authors demonstrated that injection of YLL8 did not change body weight or cause any visible side-effects. Data was missing.
12. In vitro and in vivo toxicity need to be examined.

Reviewer #1 (Remarks to the Author):

This paper reports the result of a novel screen designed to provide peptides that both bind to the surface of bone marrow stromal cells (BMSCs) and also activate the Akt pathway in doing so. The authors seem to indicate the binding is expected to be through integrin $\alpha 4\beta 1$, since the combinatorial peptide library is biased around “LDV” and QIDS” motifs, that are apparently known to be present in $\alpha 4\beta 1$ -binding peptides. The best peptide hits are then shown to stimulate some aspects of bone cell metabolism and development in vitro and in vivo.

The screen described is a significant extension of a platform described by the Lam lab previously for scoring not only cell binding to a bead-displayed ligand, but demanding a functional output of that binding event as well. Previously, a rather crude experiment was published in which peptides that induce cell death in a cancer line upon binding the bead were identified. This is study, in which a particular signaling pathway is activated, is much more appealing. Also, the peptides isolated in the original study were not at all potent, but the ones described here appear to have reasonable activity (but see point 5 below).

Overall, this paper is a novel contribution to the development of new screening technology and appears to have provided some interesting lead peptides with good activity in promoting bone formation.

The paper does have some weaknesses. It is not very well written. Sections are difficult to follow and the introduction seems to have a lot of extraneous information. There are many gaps in explaining what was done and why (see below), including why activation of the akt pathway was set as a goal in the screen. True, if one were an expert in the topic, some of this would be understood, but a general reader will quickly be lost.

Response: thank you for your overall positive comments. Yes, the screen method was adapted from the method reported by Lam KS et al. for screening of cancer drugs. We chose to use Akt activation as a screening target because Akt is known as a pro-survival signaling pathway and also was one of the most significantly upregulated kinase when we cultured MSC towards osteogenic differentiation.

We have edited and revised the manuscript significantly so that it is easier to follow, including adding explanations on what was done and why. We have also included a track-change version of the revised manuscript so that you can see our revisions. Here are the point-to-point responses to your concerns:

1. Please provide an indication of the composition of the peptide library in the main text. There is a non-peptide, urea-containing moiety on the N-terminal end. How did this come about? What was the size of the library, and what amino acids were employed in its construction?

Response: The following sentence was added to the main text “The OBOC library has four diversities with a permutation of 185,760 containing 43, 30, 8 and 18 amino acids at position X₁, X₂, X₃, and X₄, respectively. The urea-containing building block, which was originally developed by Dr. Adams group in 1998, contributed significantly to the binding potency. A ligand named BIO1211 showed a 70 pM (*K_D*) binding affinity. A reference was cited for the method we used:

(Lin, K.-C., Ateeq, H. S., Lee, W.-C., Zimmerman C. N., Castro, A., Hammond, C., Kalkunte, S., Chen, L. L., Pepinsky, R. B., Leone, D. R., Sprague, A. G., Abraham, W. M., Gill, A., Lobb, R. R., and Adams, S. P. (1998) *J. Med. Chem.*, 42, 920–934)

For structure information, we have added the following details in Supplementary Tables S1-S4:

Table S1. Forty-three amino acids for positions X₁

#1 Fmoc-Orn(Boc)-OH 	#2 Fmoc-HoSer(Trt)-OH 	#3 Fmoc-Acp-OH 	#4 Fmoc-L-HoCit
#5 Fmoc-Hyp(tBu)-OH 	#6 Fmoc-Aad(OtBu)-OH 	#7 Fmoc-D-3-Pal-OH 	#8 Fmoc-L-Phg-OH
#9 Fmoc-Nva-OH 	#10 Fmoc-Dpr(Boc)-OH 	#11 Fmoc-D-Tyr(Me)-OH 	#12 Fmoc-Aib-OH

#13 Fmoc-D-Chg-OH 	#14 Fmoc-4-Apc(Boc)-OH 	#15 Fmoc-Phe(4-Me)-OH 	#16 Fmoc-Nle-OH #17 Fmoc-D-Phe(3-Cl)-OH 	#18 Fmoc-D-HoPhe-OH 	#19 Fmoc-Aic-OH 	#20 Fmpc-Cha-OH #21 Fmoc-D-2-Nal-OH 	#22 Fmoc-L-1-Nal-OH 	#23 Fmoc-Phe(3,4-diCl)-OH 	#24 Fmoc-Bpa-OH #25 Fmoc-D-Ala-OH 	#26 Fmoc-D-Glu(OtBu)-OH 	#27 Fmoc-D-Asn(Trt)-OH 	#28 Fmoc-Gln(Trt)-OH #29 Fmoc-Ile-OH 	#30 Fmoc-D-Leu-OH 	#31 Fmoc-D-Lys(Boc)-OH 	#32 Fmoc-D-Ser(tBu)-OH #33 Fmoc-D-Met-OH 	#34 Fmoc-D-Phe-OH 	#35 Fmoc-D-Pro-OH 	#36 Fmoc-Thr(tBu)-OH #37 Fmoc-Val-OH 	#38 Fmoc-D-Trp(Boc)-OH 	#39 Fmoc-Tyr(tBu)-OH 	#40 Fmoc-Asp(OtBu)-OH #41 Fmoc-Arg(Pmc)-OH 	#42 Fmoc-D-His(Trt)-OH 	#43 Fmoc-Gly-OH 	

Table S2. Thirty amino acids for position X₂

#1 Fmoc-Ile-OH 	#2 Fmoc-D-Ala-OH 	#3 Fmoc-Abu-OH 	#4 Fmoc-D-Leu-OH #5 Fmoc-D-Pra-OH 	#6 Fmoc-Chg-OH 	#7 Fmoc-Phg-OH 	#8 Fmoc-Nva-OH #9 Fmoc-Cha-OH 	#10 Fmoc-D-Tyr(tBu)-OH 	#11 Fmoc-Asp(OtBu)-OH 	#12 Fmoc-D-Val-OH #13 Fmoc-Acpc-OH 	#14 Fmoc-Glu(OtBu)-OH 	#15 Fmoc-Ser(tBu)-OH 	#16 Fmoc-Nle-OH #17 Fmoc-Bpa-OH 	#18 Fmoc-D-2-Nal-OH 	#19 Fmoc-D-Trp(Boc)-OH 	#20 Fmoc-Ana-OH #21 Fmoc-HoSer(tBu)-OH 	#22 Fmoc-Ach-OH 	#23 Fmoc-Aad(OtBu)-OH 	#24 Fmoc-D-Thi-OH #25 Fmoc-Phe(4-Me)-OH 	#26 Fmoc-Aic-OH 	#27 Fmoc-D-Phe-OH 	#28 Fmoc-HoPhe-OH #29 Fmoc-D-Phe(3-Cl)-OH 	#30 Fmoc-D-Tyr(Me)-OH 		

Table S3. Eight amino acids for position X₃

#11 Fmoc-Asp(OtBu)-OH 	#2 Fmoc-Glu(OtBu)-OH 	#3 Fmoc-Aad(OtBu)-OH 	#4 Fmoc-Bmc(OtBu)-OH #5 Fmoc-Ile-OH 	#6 Fmoc-N-Me-Ile-OH 	#7 Fmoc-Leu-OH 	#8 Fmoc-Nle-OH 
Table S4. Eighteen building blocks for position X₄

#1 Fmoc-Ile-OH 	#2 Fmoc-Cha-OH 	#3 Fmoc-HoPhe-OH 	#4 Fmoc-Leu-OH #5 Fmoc-Nle-OH 	#6 Fmoc-N-Me-Nle-OH 	#7 Fmoc-N-Me-Ile-OH 	#8 Fmoc-HoArg(Pbf)-OH #9 Fmoc-Gln(Trt)-OH 	#10 Fmoc-Aup-OH 	#11 Fmoc-Phe(4-CF₃)-OH 	#12 Fmoc-Cpa-OH #13 Fmoc-Orn(pyra)-OH 	#14 Fmoc-Phe(3,5-diF)-OH 	#15 Fmoc-HoCit-OH 	#16 Fmoc-Cit-OH #17 Fmoc-K(A38)-OH 	#18 Fmoc-K(A12)-OH 		

2. Please provide references for the LDV and QIDS preference of the $\alpha 4\beta 1$ receptor.

Response: Two references were added to support our screening method:

The two known binding peptide motifs for $\alpha 4\beta 1$ integrin are “LDV” (Ref. Chen, L. L., Lobb, R. R., Cuervo, J. H., Adams, S. P., and Pepinsky, R. B. (1998) *Biochemistry* 37, 8743–8753) and “QIDS” (Wang, J.-H., Pepinsky, B., Stehle, T., Liu, J.-H., Karpusas, M., Browning, B., and Osborn, L. (1995) *Proc. Natl. Acad. Sci. U. S. A.* 92, 5714–5718).

3. Please show a simple picture (dark field image) of the bead-cell complex that corresponds to the fluorescence micrograph shown in Fig. 1a.

Response: We have added a dark-field image of the bead to cell binding that corresponds to the fluorescence micrograph shown in Fig. 1a.

Revised Figure 1:

Discovery a cell signaling activator with affinity for osteoprogenitor cells

- a** 1, screened focus library with BMSCs
2, searched beads with both binding and positive for p-Akt activation

- b** 1, sequencing the beads
2, re-synthesized leading bead library
3, binding affinity to "osterix"+ cells

- c** Re-confirmed activation of Akt on cells upon binding to beads

- d** 1, lead peptides that activated p-Akt upon binding
2, with affinity to osteoprogenitor cells

YLL3 ↓ YLL8

Schematic for the discovery of osteogenic specific peptides. **(a)** Bone marrow stromal cells (BMSCs) were obtained from mice and maintained in mesenchymal maintenance medium for two weeks (~3 passages), switched to osteogenic medium for 3 to 5 days, then incubated with a focused integrin library for 1 hour. These bead-bound osteoprogenitor cells (OPCs) were then fixed and stained for phosphorylated Akt. Positive beads were identified as displaying both cell binding to the beads and positive p-Akt expression in the cells bound to the beads. **(b)** Positive beads from the first screen were identified, sequenced, and resynthesized. The beads were incubated with osterix+ red cells to identify beads with a high affinity towards osterix+ cells semi-quantitatively. **(c)** Positive beads from the second screen were resynthesized and re-incubated with OPCs to confirm p-Akt activation. P-Akt was stained green. Some cells on beads were positive for p-Akt. Cells not attached to beads were p-Akt negative. **(d)** From the described screening process, YLL3 and YLL8 were selected for decoding and resynthesized for further characterization.

4. The text refers to Fig. 2e, but there is no panel e in Fig. 2.

Response: We apologize for the oversight. We have a new Fig. 2a-d and the previous reference to Fig. 2e is updated to Fig. 2h.

Revised Figure 2:

Osteogenic peptides YLL3 and YLL8 binding affinities towards osteoprogenitor cells and *in vitro* osteogenic differentiation property. BMSCs obtained from mouse or human were plated on collagen-I coated wells and incubated with auto-fluorescence-quenched beads displaying YLL3 or YLL8 in osteogenic media. (a) Quantitative measurement of alkaline phosphatase (ALP) at day 7 and alizarin red (AR) levels at day 21 in human-derived cell cultures treated with YLL3 or YLL8. (b) Representative images of binding affinities with human-derived MSCs after co-cultured with beads displaying YLL3 or YLL8 at 12 hours. (c) Representative images of ALP staining in human-derived MSCs cultured with beads displaying YLL3 or YLL8 for 14 days. (d) Human-derived MSCs were cultured with YLL3, and YLL8 peptides at indicated concentration in osteogenic media for 10 days and stained for ALP. (e) Quantitative measurement of ALP at day 7 and AR at day 21 in mouse-derived cell cultures. (f) Representative images of ALP staining in mouse BMSCs cultured with beads displaying YLL3 or YLL8 at day 10. (g) Representative images of AR staining in mouse BMSCs cultured with beads displaying YLL3 or YLL8 at day 21. (h) Mouse BMSCs were cultured with YLL3 and YLL8 peptides at 6 x 10⁻⁸ M in osteogenic media for 10 or 21 days to measure ALP levels at day 10, and AR levels at day 21. *, p < 0.05 vs. Control.

5. It would good to see a dose-response of the induction of the Akt pathway by the selected peptides. From the data presented in the paper, it is not clear how potent the peptide is.

Response: We have performed new dose-response studies and demonstrated that the phosphorylation level of Akt during human MSC osteogenic differentiation was indeed dependent on the concentration of YLL3 and YLL8 used in the experiment. The new data is provided in the new Figure. 3 d-e. The new data showed both YLL3 and YLL8 activated p-Akt at a higher concentration at 10⁻⁶M than 10⁻⁸M.

(d) Human-derived MSCs were cultured with YLL3 or YLL8 at indicated concentration in osteogenic media for two hours. Cell lysates were collected for western blotting of p-Akt, Akt, and beta-actin. (e) Representative bands and quantitative pixel density from n=3 replicates.

6. Fig. 3 is difficult to follow. It would be so much better just to provide the grid in Fig. 3b. rather than having Fig. 3a be a separate legend of sorts. Also, please explain the significance of dots being there or not. There are red boxes in Fig. 3B, but their significance is not explained.
 Response: We have reorganized Fig. 3 so that it is more reader-friendly. We revised the corresponding descriptions for the "dots" and clarified that the "red rectangles" indicated higher expressed proteins as compared to the control-treated group. We also revise the figure legends

so that it is easier to follow. Thanks for bringing this out to us!
 Revised Figure 3:

YLLs activated the Akt signaling. (a) Mouse-derived MSCs were cultured with YLL3 or YLL8 peptides or hPTH (1-34) at 6 x 10⁻⁸M in osteogenic medium for 2 hours. The cell lysates were collected and used in an Akt protein array, measuring 18 phosphorylated proteins within the Akt signaling pathway. (b) The corresponding protein mapping of the Akt array listed in (a). (c) Quantitative pixel density/blank control from selected dots highlighted in rectangles in b, which were differentially expressed compared to the control. (d) Human-derived MSCs were cultured with YLL3 or YLL8 at indicated concentration in osteogenic media for two hours. Cell lysates were collected for western blotting of p-Akt, Akt, and beta-actin. Representative bands and (e) quantitative pixel density from n=3 replicates.

Reviewer #2 (Remarks to the Author):

This study developed a screening method for identifying osteoprogenitor-specific peptides. The authors developed it as an “one-entity tri-action” screening principle. Two targeting peptides were screened, and defined as YLL3 and YLL8. In vitro and in vivo study showed that they could increase osteogenic differentiation and osteoblastic activity. They also conjugated the two targeting peptides with bisphosphonate, which sustained bone anabolic action but lower drug dosing frequency. The work was original and show potential for translational application. However, the following concerns limited the novelty of this study and several critical evidences were missing in this manuscript.

Response: Thank you for your comments and have addressed your concerns that you listed below that might limited the novelty of this study included performing additional efficacy studies on hormonal deficiency models. We showed particularly one peptide, YLL3, prevented bone loss in multiple skeletal sites via increasing bone formation in both females and males (new Figure 5). Here are the point-to-point answers to your concerns:

1. Definition of osteoprogenitor was not clear. In Figure 1, MSCs were induced for three days and then used for screening. At day 3, how many MSCs were transformed to osteoprogenitors was not clear. The authors used osterix as a marker for osteoprogenitors and showed good binding affinity of the screened peptides with osteoprogenitors rather than lymphocytes. Whether the screened peptides could recognize MSCs need to be confirmed.

Response: Yes, we did not directly We did not directly measure the binding affinities of our peptides for the MSCs during the screening step. Instead, we used semiquantitative methods, i.e., the binding affinity of beads displaying different peptides with osterix + cells that were extracted from bone marrow cells of osterix-mCherry mice. Osterix is a marker for early osteochondral progenitor cells (refs 34, 35). Our screening method was to identify peptides (i) with high-affinity towards the osteoprogenitor cells, and (ii) that induce Akt signaling pathway. We did not directly measure the binding affinities of our peptides for the MSCs during the screening step. Figure 1b showed representative semi-quantitative methods we used as part of the screening method.

Colony-forming assay through measurement of ALP is an approach that is often used to monitor MSC to osteogenic differentiation. ALP usually starts to increase from day 3 in osteogenic culture condition and peaks at days 7-10. Since we would like to target early osteoprogenitor cells and not the mature osteoblast, we used a three-day protocol to culture MSC in osteogenic culture media before the cells were used for screening. Here are two representative references we used for the osteogenesis culture:

Langenbach, F. & Handschel, J. Effects of dexamethasone, ascorbic acid, and beta-glycerophosphate on the osteogenic differentiation of stem cells in vitro. *Stem Cell Res Ther* **4**, 117, DOI:10.1186/scrt328 (2013).

Guan, M. *et al.* Directing mesenchymal stem cells to bone to augment bone formation and increase bone mass. *Nat Med* **18**, 456-462, DOI:10.1038/nm.2665 (2012).

Additionally, we have repeated the studies using human MSCs that were cultured in osteogenic medium for three days before they were used for the studies, which we presented this new data as a new Figure 2 a-d. Our results using human MSC suggested similar binding affinity towards

osteoprogenitor cells as well as enhancing the human MSC towards osteogenic differentiations when the cells were co-cultured with the “osteogenic” peptides that we developed.

2. Evidences from this study showed that the screened peptides activated AKT signaling in osteoprogenitors. What is the mechanism for the agonistic action of the screened peptides? Is the mechanism same for the screened YLL3 and YLL8?

Response: Similar screening methods were used to screen YLL3 and YLL8, both demonstrated similar binding affinities towards osterix+ cells. We added data on YLL3 and YLL8 on the activation of Akt signaling in human-derived OPCs (Figure 3 d-e) that showed the ability to stimulate osteogenesis through increasing bone mineral apposition rate, a surrogate for osteoblast activities, was higher in YLL3-treated cells *in vitro*. YLL3 induced higher bone formation in the hormonal deficiency model (new Figure 5) as well as in the fracture healing model (Figure 6).

New Figure 5
YLLs prevented bone loss in gonadal-deficient mice. Two-month-old C57BL/6 mice were ovariectomized (female, OVX) or orchietomized (male, ORX) and treated with PBS, YLL3 or YLL8 at 10 $\mu\text{g}/\text{kg}$ or hPTH (1-34) at 40 $\mu\text{g}/\text{kg}$, sc., 5x/week for 28 days (n=6/group). All mice received Calcein and Alizarin red at -8 and -1 days before euthanization. (a) Quantitative measurements of trabecular bone microstructure at the proximal tibial metaphysis (PTM). (b) Representative 3D images of the PTM from the female mice in the OVX study. (c) Trabecular bone surface-based bone formation was measured in the PTM in the OVX mice. (d) Representative fluorescence images from the PTM trabecular regions in the OVX study. (e) Trabecular bone surface-based bone formation was measured in the lumbar vertebral bodies in the ORX mice. *, $p < 0.05$ vs. PBS-treated OVX group.

Revised Figure 6.

YLL3 induced higher callus mineralization during fracture healing. Female Prx1^{ERT}-GFP mice were fractured at 2 months of age and received tamoxifen at 10 mg/kg for three days. YLL3 or YLL8 were given at 10 µg/kg, and hPTH (1-34) was given at 40 µg/kg, s.c., 5x/week for 10 or 21 days. (a) Representative callus images from the outer edge of the callus in indicated treatment groups at day 10 post-fracture. (b) Representative microCT thickness mapping of the callus, callus bone volume, and bone mineral content from indicated treatment groups at day 10 post-fracture. (c) Representative callus images from the outer edge of the callus in indicated treatment groups at 21 days post-fracture. Alizarin red was given at -1 day before euthanization (d) Representative microCT thickness mapping of the callus, callus bone volume, and bone mineral content from indicated treatment groups at day 21 post-fracture.

3. In Figure 5, in vivo data showed that the two peptides (YLL3 and YLL8) could increase mineral apposition rate, bone formation, bone mass, and bone strength in young mice as well as expedited fracture repair. Could they directly recognize bone and osteoprogenitors with conjugating Aln? What is the rationale for conjugating YLL3 and YLL8 with Aln?

Response: Conjugating Aln to YLL3 and YLL8 allows (i) delivery and retention of higher concentration of the peptides to the bone, and (ii) capturing of osteoprogenitor cells by the immobilized peptides at the bone matrix. We thought we could further increase their bone affinity by conjugating the peptide to alendronate, a bone-targeting drug using a similar method reported by Guan et al. : Guan, M. *et al.* Directing mesenchymal stem cells to the bone to augment bone formation and increase bone mass. *Nat Med* **18**, 456-462, doi:10.1038/nm.2665 (2012). We also added a new supplementary Figure 6 to show the bone affinity of the peptide-Aln conjugation.

However, YLL3 lost its anabolic effect after the conjugation with Aln. Work is currently underway to synthesize a cleavable YLL3-Aln conjugate via an ester bond (e.g., through the side chain of Asp or Ser in YLL3). We have revised discussion as followings:

We conjugated the YLLs to a bone targeting drug, alendronate (Aln). In this case, Aln was used as a “carrier” for bone-specific delivery of the peptide. The bisphosphonates (BPs) are known to contain two phosphonate groups with covalently bonded side chains known as R₁ and R₂. The presence of a hydroxyl (OH) in the R₁ chain gives an affinity to the bone. The anti-resorptive potency of BPs is believed to reside in the R₂ chain, especially for the N-containing BPs⁷⁴. In our peptide-BP conjugation, we used the NH₂ in the R₂ chain to link Aln to the peptide. The published studies using similar methods also did not observe anti-resorptive effects from these peptide-BP conjugation^{21-23,75,76}. This Aln-conjugation was deposited to the skeleton within 6 hours. After 7 days, the drug remained primarily in the skeleton²². To further determine the enhancing bone-targeted effect of YLLs, we performed an *in vivo* experiment and demonstrated 2s.c. doses of YLL8-Aln increased both trabecular and cortical bone mass. Collectively, these studies demonstrated that YLL8 could stimulate osteoblast differentiation *in vitro* and *in vivo*, demonstrating the efficacy of YLL8-Aln on bone formation. Dynamic bone histomorphometric and histologic analyses confirmed that YLL8 significantly promoted bone formation via enhancing the osteoblast activity. Surprisingly, YLL3-Aln lost its anabolic effects on bone when it was conjugated to Aln, which might due to the conjugation change the chemical configuration of YLL3 necessary for bone formation, or higher dose/dosing frequency may be needed to achieve the anabolic effects observed for the peptide itself. Further dose-defining studies may require to better define the potential anabolic action for YLL3-Aln. Alternatively, we can modify our conjugation method so that the YLL3 can be cleavable from the Aln conjugation so that the free peptide could be released after it reaches the bone.

New Supplementary Figure 6:

Near-infrared fluorescence imaging of the tibia after the mice were injected with YLL8-Aln-DiD for 4 hours. Blue arrows indicated bone homing of the injected drug in bright white.

4. **In vivo biophotonic imaging and Immunofluorescent staining data need to be provided to show the bone and osteoprogenitors targeting.**

Response: We have provided the biophotonic imaging of homing/affinity of to YLLs to the bone surface after injection in supplemental Figure 6.

In the fracture study, we used an inducible Prx1-GFP mouse so that we could monitor the effects of YLLs on the activation of the endogenous osteoprogenitor cells (Prx1+, in green) during fracture and the relationship with bone formation (red fluorescence). The data is updated in Figure 6. We used a double-fluorescence labeling technique, a well-accepted technique to label osteogenesis *in vivo*, in all the three *in vivo* studies (intact, OVX, ORX, and fracture) and showed the peptides increase mineral apposition rate and bone formation, surrogates for osteoprogenitor targeting.

5. The authors suggested that YLL3 and YLL8 were pro-survival ligands which stimulated osteogenesis via enhancing the AKT signaling, it is important to provide evidences that YLL3 and YLL8 could enhance proliferation and survival of the osteoprogenitors, which were missing in this manuscript.

Response: We have added the dose-response study of the YLL3 and YLL8, on cell proliferation/survival (MTS experiment), as well as on Akt activation (western blots) using human-derived MSC: Figure 3d-e and supplemental Figure 2, suggested that the peptides might enhance cell survival.

We also observed higher mineral apposition rates in intact, gonadal deficient and fractured mice, suggested higher osteoblast activities *in vivo*, indirect evidence of prolonged osteoblast life span and activities.

6. In Figure 6, YLL3 and YLL8 could be chosen as controls to show whether the conjugation of Aln compromise the anabolic action of YLL3 and YLL8.

Response: data for YLL3 and YLL8 as “controls” were added to the new Figure 7. Alendronate was used as a bone-targeting agent (1/4 molar of the conjugates, about 150ug total exposure). Alendronate-conjugated YLL3 did show compromised anabolic action. We added discussions on this finding as provided above under #3 comments.

Revised Figure 7. YLL8-Aln increased trabecular and cortical bone mass. Two-month-old female mice were treated with PBS control, YLL3 or YLL8 at 10 $\mu\text{g}/\text{kg}$ or YLL3-Aln or YLL8-Aln at 300 $\mu\text{g}/\text{kg}$ s.c., every other week for 1 month. Mice were euthanized at day 28 (n=5-7/group). (a) Quantitative distal femoral trabecular and middle-femur cortical bone volume measured by microCT. (b) Representative 3D images of the trabecular region taken from the distal femurs in indicated treatment groups. (c) Representative images of the middle femoral cortex in indicated treatment groups. *, $p < 0.05$ vs. PBS.

7. For the animal models, the authors chose young mice and bone fracture mice, it is critical to examine the bone anabolic action in aged osteoporotic mice or OVX mouse models.

Response: We have now provided new data on the hormone-deficiency induced bone loss models. New data is presented in a new Figure 5 (please refer to #2 comment above) shows both YLL3 and YLL8, especially YLL3, were effective in preventing bone loss following gonadal deficiency.

8. The authors used mouse MSCs when screening peptides. Binding affinity of the screened peptides with human or rat cells should be tested to confirm the translational potential of the peptides.

Response: We have repeated the binding study and the osteogenic studies using human MSCs. New data is provided in Figure 2a-d and Figure 3d-e (please refer to the new Figures 2a-d, Figures 3d-e above). A study using human MSC showed similar results to the ones we previously had using mouse cells. Thanks for the suggestion!

9. In Figure 1, the authors demonstrated that they identified 22 peptides and focused on two leading peptides (YLL3 and YLL8, Fig. 1d) based on their high binding affinity to osterix cells as well as their osteogenic effects in vitro. However, evidences of their osteogenic effects were missing.

Response: We add the selected osteogenic effects of other peptides we had from the screening using ALP readings, including those from YLL3 and YLL8, as a new supplemental Figure 2a.

Supplementary Figure 2a. ALP levels of select beads displaying peptides cultured with mouse MSC for 7 and 14 days.

10. Figure 2d was missing.

Response: We apologize for the oversight. We have revised Figure 2 and updated the figure legend please refer to the new Figure 2 and its figure legend above).

11. In Figure 4, the authors demonstrated that injection of YLL8 did not change body weight or cause any visible side-effects. Data was missing.

Response: Bodyweight data are now provided in Supplemental Figures 3c-e. These data also serve as a surrogate for toxicity.

a

b

c

Body weights
Daily injection of YLL3 or YLL8 for 4 weeks

d

Body weights (OVXed mice)
Daily injection of YLL3 or YLL8 for 4 weeks

e

Body weights (ORXed mice)
Daily injection of YLL3 or YLL8 for 4 weeks

New Supplemental Figure 3. (a) MTS results for YLL3 and (b) for YLL8 after cultured with human MSC for up to three days in indicated concentrations. (c) Bodyweight changes in the experiment when YLL3 and YLL8 were given daily to intact adult female mice for 4 weeks. (d) Bodyweight changes in the experiment when YLL3 and YLL8 were given daily to ovariectomized mice for four weeks. (e) Bodyweight changes in the experiment when YLL3 and YLL8 were given daily to orchietomized mice for four weeks.

12. In vitro and in vivo toxicity need to be examined.

Response: In vitro toxicity study was done, and data is now provided in Supplemental Figure 3a. Bodyweight data are now provided in Supplemental Figure 3b. These data also serve as a surrogate for toxicity. The vital organs (heart, lung, liver, kidney, and spleen) from the gonadal deficiency-study were processed for pathology, and we did not detect organ toxicity at the doses we used. We are also in the process of obtaining funding to perform a single and a repeated dose toxicity study with a CRO under GLP condition. The data is useful for IND application moving forward but is out of scope under the current investigation. Thank you so much for your comments and suggestions!

Reviewers' Comments:

Reviewer #1:

Remarks to the Author:

The authors have revised the paper quite extensively in response to the reviewer comments. It is now far easier to read, which was a major problem previously. Also, the authors have addressed all of the important experimental points that were raised. I think the paper is now acceptable.

Reviewer #2:

Remarks to the Author:

My concerns are well addressed by the authors. Answers from the authors significantly improve the quality and logic of this manuscript. I am glad to know that the authors is considering the IDN application, which is very important for the translational medicine of this study.